# DAFA: DISTANCE-AWARE FAIR ADVERSARIAL TRAINING

Hyungyu Lee[1], Saehyung Lee[1], Hyemi Jang[1], Junsung Park[1], Ho Bae[2,*], and Sungroh Yoon[1,3,*]

[1]Electrical and Computer Engineering, Seoul National University
[2]Department of Cyber Security, Ewha Womans University
[3]Interdisciplinary Program in Artificial Intelligence, Seoul National University
rucy74@snu.ac.kr    halo8218@snu.ac.kr    wkdal9512@snu.ac.kr
jerryray@snu.ac.kr    hobae@ewha.ac.kr    sryoon@snu.ac.kr

## ABSTRACT

The disparity in accuracy between classes in standard training is amplified during adversarial training, a phenomenon termed the robust fairness problem. Existing methodologies aimed to enhance robust fairness by sacrificing the model's performance on easier classes in order to improve its performance on harder ones. However, we observe that under adversarial attacks, the majority of the model's predictions for samples from the worst class are biased towards classes similar to the worst class, rather than towards the easy classes. Through theoretical and empirical analysis, we demonstrate that robust fairness deteriorates as the distance between classes decreases. Motivated by these insights, we introduce the Distance-Aware Fair Adversarial training (DAFA) methodology, which addresses robust fairness by taking into account the similarities between classes. Specifically, our method assigns distinct loss weights and adversarial margins to each class and adjusts them to encourage a trade-off in robustness among similar classes. Experimental results across various datasets demonstrate that our method not only maintains average robust accuracy but also significantly improves the worst robust accuracy, indicating a marked improvement in robust fairness compared to existing methods.

## 1 INTRODUCTION

Recent studies have revealed the issue of accuracy imbalance among classes (He & Garcia, 2009). This imbalance becomes even more pronounced during adversarial training, which utilizes adversarial examples (Szegedy et al., 2013) to enhance the robustness of the model (Madry et al., 2017). This phenomenon is commonly referred to as "robust fairness problem" (Xu et al., 2021). Existing research has introduced methods inspired by long-tailed (LT) classification studies (He & Garcia, 2009; Zhang et al., 2023) to mitigate the challenge of achieving robust fairness. LT classification tasks tackle the problem of accuracy imbalance among classes, stemming from classifier bias toward classes with a substantial number of samples (head classes) within the LT dataset. The methods proposed for LT classification mainly apply opposing strategies to head classes and tail classes–those classes within LT datasets that have a limited number of samples. For instance, methods proposed by Cao et al. (2019); Khan et al. (2019); Menon et al. (2020) deliberately reduce the model output for head classes while augmenting the output for tail classes by adding constants. These approaches typically lead to improved accuracy for tail classes at the expense of reduced accuracy for head classes.

Benz et al. (2021) noted similarities between the fairness issue in LT classification and that in adversarial training. They corresponded the head and tail classes in LT classification with the easy and hard classes in adversarial training, respectively. Employing a similar approach, they addressed robust fairness by compromising the model's relatively higher robustness on easy classes to bolster the robustness on hard classes. Subsequent studies (Wei et al., 2023; Li & Liu, 2023) also addressed robust fairness by applying stronger regularization to the learning of easy classes compared to that of hard classes.

---

*Corresponding authors

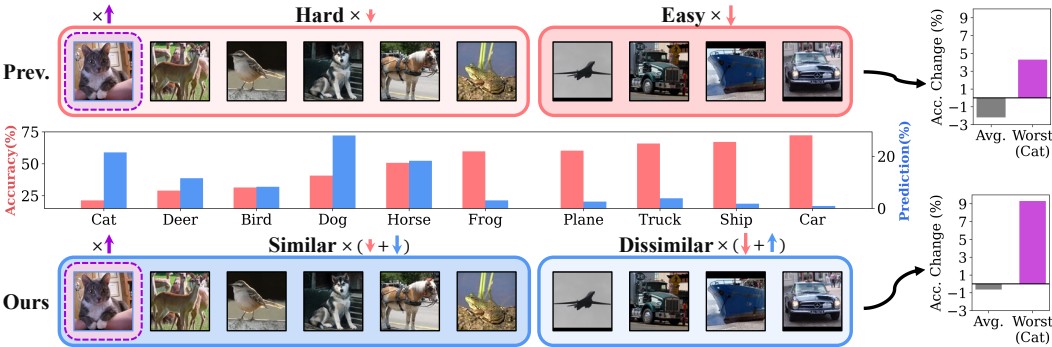

Figure 1: The figure illustrates the performance improvement for the worst class (cat) when training the robust classifier (Zhang et al., 2019) on the CIFAR-10 dataset (Krizhevsky et al., 2009), by varying the training intensity for different class sets. While the previous approach (Xu et al., 2021) limits the training of each class proportional to class-wise accuracy, our method considers inter-class similarity. Our approach intensifies the training constraints on animal classes, which are neighbors to the cat class, compared to previous methods, while relaxing constraints on non-animal classes.

However, in this study, we find that while the fairness issue in LT classification is caused by biases towards head classes, the fairness issue in adversarial training is caused by biases towards similar classes. Fig. 1 contrasts the outcomes using conventional robust fairness improvement techniques versus our method. The figure, at its center, depicts the accuracy of each class (red) on the CIFAR-10 dataset alongside the prediction of the worst class samples, "cat" (blue). The prediction of the cat class highlights a significantly higher likelihood of misclassification towards similar classes, such as the dog class, compared to dissimilar classes. In light of these findings, our strategy addresses robust fairness from the perspective of "inter-class similarity". Existing methods majorly constrain the learning of easy (non-animal) classes to boost the worst robust accuracy. In contrast, our method restricts the learning of similar (animal) classes more than previous methods while imposing fewer constraints on the learning of dissimilar (non-animal) classes. The results in Fig. 1 indicate that our method notably enhances the worst class's performance while maintaining average performance compared to the previous method. This underscores the idea that in adversarial training, performance-diminishing factors for hard classes are attributed to similar classes. Therefore, limiting the learning of these neighboring classes can serve as an effective means to improve robust fairness.

In our study, we conduct a thorough investigation into robust fairness, considering inter-class similarity. We theoretically and empirically demonstrate that robust fairness deteriorates by similar classes. Building on these analyses, we introduce the Distance-Aware Fair Adversarial training (DAFA) method, which integrates inter-class similarities to mitigate fairness concerns in robustness. Our proposed method quantifies the similarity between classes in a dataset by utilizing the model output prediction probability. By constraining the learning of classes close to hard classes, it efficiently and substantially enhances the performance of the worst class while preserving average accuracy. Experiments across multiple datasets validate that our approach effectively boosts the worst robust accuracy. To summarize, our contributions are:

- We conduct both theoretical and empirical analyses of robust fairness, taking into account inter-class similarity. Our findings reveal that in adversarial training, robust fairness deteriorates by their neighboring classes.

- We introduce a novel approach to improve robust fairness, termed Distance-Aware Fair Adversarial training (DAFA). In DAFA, to enhance the robustness of hard classes, the method allocates appropriate learning weights and adversarial margins for each class based on inter-class distances during the training process.

- Through experiments on various datasets and model architectures, our method demonstrates significant enhancement in worst and average robust accuracy compared to existing approaches. Our code is available at https://github.com/rucy74/DAFA.

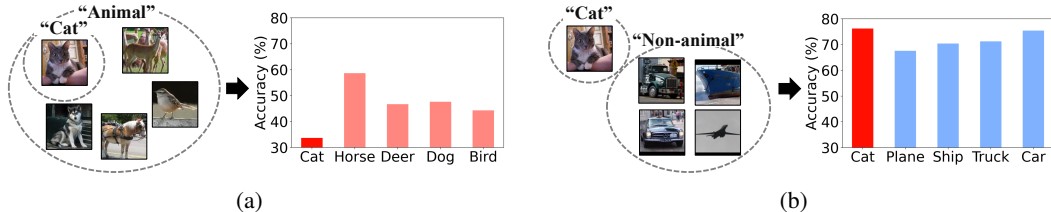

(a)                                                                 (b)

Figure 2: (a) and (b) depict the results from two distinct 5-class classification tasks. (a) presents the robust accuracy of the robust classifier trained on the cat class alongside other animal classes from CIFAR-10, while (b) illustrates the robust accuracy of the robust classifier trained on the cat class in conjunction with non-animal classes in CIFAR-10.

## 2 PRELIMINARY

Adversarial training, as described in Madry et al. (2017), uses adversarial examples within the training dataset to improve robustness. An alternative approach to adversarial training is the TRADES method (Zhang et al., 2019). This method modulates the balance between model robustness and clean accuracy by segregating the training loss into the loss from clean examples and a KL divergence term to foster robustness. Consider a dataset denoted by $\mathcal{D} = \{(\boldsymbol{x}_i, y_i)\}_{i=1}^{n}$. Here, $\boldsymbol{x}_i \in \mathbb{R}^d$ is clean data, and each $y_i$ from the class set indicates its corresponding label. The equation of the TRADES method can be expressed as the subsequent optimization framework:

$$\min_{\theta} \sum_{i=1}^{n} \big( \mathcal{L}_{CE}(f_\theta(\boldsymbol{x}_i), y_i) + \beta \cdot \max_{\|\boldsymbol{\delta}_i\| \leq \epsilon} \mathrm{KL}(f_\theta(\boldsymbol{x}_i), f_\theta(\boldsymbol{x}_i + \boldsymbol{\delta}_i)) \big). \tag{1}$$

Here, $\theta$ denotes the parameters of the model $f$. The cross-entropy loss function is represented by $\mathcal{L}_{CE}$. $\mathrm{KL}(\cdot)$ represents KL divergence, and $\beta$ is a parameter that adjusts the ratio between the clean example loss and the KL divergence term. Additionally, $\epsilon$ is termed the adversarial margin. This margin sets the maximum allowable limit for adversarial perturbations. Regarding adversarial attacks, the fast gradient sign method (FGSM) stands as a prevalent technique for producing adversarial samples (Goodfellow et al., 2014). Projected gradient descent (PGD) can be seen as an extended iteration of FGSM (Madry et al., 2017). Adaptive auto attack ($A^3$) is an efficient and reliable attack mechanism that incorporates adaptive initialization and statistics-based discarding strategy (Liu et al., 2022). Primarily, we use the TRADES methodology and $A^3$ as our adversarial training and evaluation methods, respectively.

## 3 ANALYSIS

In classification tasks, the difficulty of each class can be gauged by the variance between samples within the class or its similarity to other classes, i.e., the inter-class distance. Specifically, a class with a substantial variance among samples poses challenges for the classifier, as it necessitates the recognition of a broad spectrum of variations within that class. Previous studies utilized a class's variance for theoretical modeling of the robust fairness issue (Xu et al., 2021; Ma et al., 2022; Wei et al., 2023). These studies demonstrated that the accuracy disparity arising from differences in class variance is more pronounced in adversarial settings compared to standard settings.

On the other hand, concerning inter-class distance, we can posit that a class is harder to train and infer if it closely resembles other classes. Fig 2a shows the results of a classification task that includes the "cat" class alongside four other animal classes, which are close or similar to the cat class, all under the same superclass: animal. Conversely, Fig 2b presents results with the cat class among four distinct non-animal classes. Notably, while the cat data remains consistent across tasks, its accuracy is lowest when grouped with similar animal classes and highest with dissimilar non-animal classes. These observations suggest that a class's difficulty can vary, making it either the easiest or hardest, based on its proximity or similarity to other classes in the dataset. Consequently, the similarity between classes, which is measured by inter-class distance, plays a pivotal role in determining the difficulty of a class.

## 3.1 THEORETICAL ANALYSIS

In this section, we conduct a theoretical analysis of robust fairness from the perspective of inter-class distance. Intuitively, a class is hard to learn and exhibits lower performance when it is close to other classes than when it is distant. Based on this intuition, we aim to demonstrate that robust fairness deteriorates when classes are in close proximity.

**A binary classification task** We define a binary classification task using Gaussian mixture data, which is similar to the binary classification task presented by Xu et al. (2021); Ma et al. (2022). Specifically, we introduce a binary classification model given by $f(\boldsymbol{x}) = \text{sign}(\boldsymbol{w}^\top \boldsymbol{x} + b)$. This model represents a mapping function $f : \mathcal{X} \to \mathcal{Y}$ from input space $\mathcal{X}$ to the output space $\mathcal{Y}$. Here, the input-output pair $(\boldsymbol{x}, y)$ belongs to $\mathbb{R}^d \times \{\pm 1\}$, with sign denoting the sign function. We define an input data distribution $\mathcal{D}$ as follows:

$$y \overset{u.a.r}{\sim} \{-1, +1\}, \ \boldsymbol{\mu} = (\eta, \eta, \cdots, \eta), \qquad \boldsymbol{x} \in \mathbb{R}^d \sim \begin{cases} \mathcal{N}(\boldsymbol{\mu}, \sigma^2 I), & \text{if } y = +1 \\ \mathcal{N}(-\alpha\boldsymbol{\mu}, I), & \text{if } y = -1 \end{cases} \qquad (2)$$

Here, $I \in \mathbb{R}^{d \times d}$ denotes the identity matrix. We assume that each class's distribution has different mean and variance, and this variability is controlled by $\sigma > 1$ and $\alpha \geq 1$. The class $y = +1$ with a relatively large variance exhibits higher difficulty and is therefore a harder class than $y = -1$. Here, we initially observe the performance variation of the hard class $y = +1$ as the distance between the two classes changes.

We define the standard error of model $f$ as $\mathcal{R}_{nat}(f) = Pr.(f(\boldsymbol{x}) \neq y)$ and the standard error for a specific class $y \in \mathcal{Y}$ as $\mathcal{R}_{nat}(f|y)$. The robust error, with a constraint $\|\boldsymbol{\delta}\| \leq \epsilon$ (where $\epsilon$ is the adversarial margin), is represented as $\mathcal{R}_{rob}(f) = Pr.(f(\boldsymbol{x} + \boldsymbol{\delta}) \neq y)$. The robust error for a class $y \in \mathcal{Y}$ is denoted by $\mathcal{R}_{rob}(f|y)$. Let $f_{nat} = \arg\min_{\boldsymbol{w},b} \mathcal{R}_{nat}$ and $f_{rob} = \arg\min_{\boldsymbol{w},b} \mathcal{R}_{rob}$. Given these definitions, the aforementioned binary classification task conforms to the following theorem:

**Theorem 1.** *Let $0 < \epsilon < \eta$ and $A \equiv \frac{2\sqrt{d}\sigma}{\sigma^2 - 1} > 0$. Given a data distribution $\mathcal{D}$ as described in equation 2, $f_{nat}$ and $f_{rob}$ exhibit the following standard and robust errors, respectively:*

$$\mathcal{R}_{nat}(f_{nat}| + 1) = Pr.\left(\mathcal{N}(0,1) < -A(\frac{1+\alpha}{2}\eta) + \sqrt{(\frac{A}{\sigma})^2(\frac{1+\alpha}{2}\eta)^2 + \frac{2\log\sigma}{\sigma^2 - 1}}\right), \qquad (3)$$

$$\mathcal{R}_{rob}(f_{rob}| + 1) = Pr.\left(\mathcal{N}(0,1) < -A(\frac{1+\alpha}{2}\eta - \epsilon) + \sqrt{(\frac{A}{\sigma})^2(\frac{1+\alpha}{2}\eta - \epsilon)^2 + \frac{2\log\sigma}{\sigma^2 - 1}}\right). \qquad (4)$$

*Consequently, both $\mathcal{R}_{nat}(f_{nat}| + 1)$ and $\mathcal{R}_{rob}(f_{rob}| + 1)$ decrease monotonically with respect to $\alpha$.*

A detailed proof of Theorem 1 can be found in Appendix C.1. Theorem 1 suggests that as $\alpha$ increases, indicating an increasing distance between the two classes, the error of the class $y = +1$ decreases under both standard and adversarial settings. Furthermore, we illustrate that the disparity in robust accuracy between the two classes also reduces when the distance between them increases.

**Theorem 2.** *Let $D(\mathcal{R}(f))$ be the disparity in errors for the two classes within dataset $\mathcal{D}$ as evaluated by a classifier $f$, Then $D(\mathcal{R}(f))$ is*

$$D(\mathcal{R}(f)) \equiv \mathcal{R}(f| + 1) - \mathcal{R}(f| - 1)$$

$$= Pr.\left(\frac{A}{\sigma}g(\alpha) - \sqrt{A^2 g^2(\alpha) + h(\sigma)} < \mathcal{N}(0,1) < -Ag(\alpha) + \sqrt{(\frac{A}{\sigma})^2 g^2(\alpha) + \frac{h(\sigma)}{\sigma^2}}\right), \qquad (5)$$

*where $A = \frac{2\sqrt{d}\sigma}{\sigma^2 - 1}$, $h(\sigma) = \frac{2\sigma^2 \log\sigma}{\sigma^2 - 1}$, and in a standard setting $g(\alpha) = \frac{1+\alpha}{2}\eta$, while in an adversarial setting, $g(\alpha) = \frac{1+\alpha}{2}\eta - \epsilon$. Then, the disparity between these two errors decreases monotonically with the increase of $\alpha$.*

Theorem 2 demonstrates that both in standard and adversarial settings, the accuracy disparity is more significant for closer classes compared to distant ones. A proof of Theorem 2 is provided in Appendix C.1.

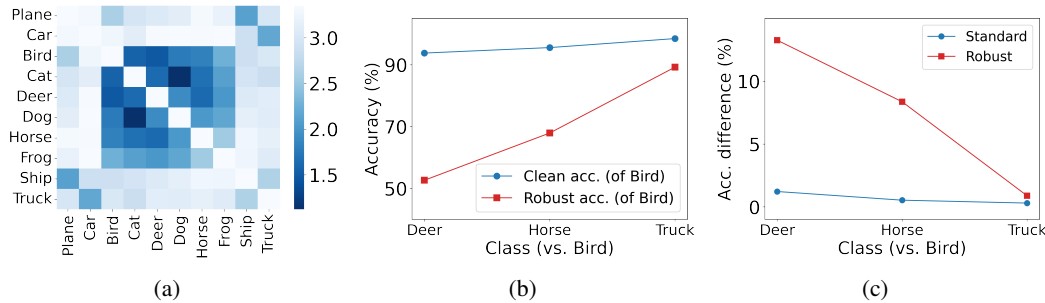

Figure 3: (a) displays the class distances between classes in CIFAR-10 as represented in the TRADES model. (b) and (c) depict the results from binary classification tasks on CIFAR-10's bird class paired with three distinct classes: deer, horse, and truck. Specifically, (b) presents the performance for the bird class, while (c) displays the accuracy disparity for each respective classifier.

## 3.2 EMPIRICAL VERIFICATION

**Empirical verification of theoretical analysis** We undertake empirical verification based on preliminary theoretical findings. We select three pairs of classes with varying distances between them from the CIFAR-10 dataset. Fig. 3a displays the distances between classes in the robust classifier. Based on this, we conduct binary classification tasks for the bird class, which is the hard class, against three other easy classes, ordered by distance: deer, horse, and truck. Initially, we compare the performance of the bird class in the standard and robust classifiers to validate the assertion from Theorem 1 that as the distance between two classes increases, the performance of the hard class increases. In Fig. 3b, we observe that as the distance to the bird class increases, both performances improve, a result that aligns with Theorem 1. Secondly, corresponding to Theorem 2, we validate that distant classes exhibit a smaller accuracy disparity. The results in Fig. 3c highlight that as the distance to the bird class increases, accuracy disparity decreases. Thus, we confirm that the theorems presented earlier are empirically verified. Further details are provided in Appendices D.2 and C.2.

**Class-wise distance** The theoretical and empirical analyses presented above indicate that a class becomes more challenging to learn when it is close to other classes. Intuitively, this suggests that in a multi-classification task, the more a class's average distance from other classes decreases, the harder it becomes. Motivated by this observation, we conduct experiments to examine the correlation between the average distance to other classes and class difficulty, aiming to determine the extent to which this average distance reflects class difficulty in an adversarial setting. For simplicity, we will refer to the average distance of a class to other classes as the 'class-wise distance'.

Fig. 4a presents the correlation between class-wise accuracy and distance. The figure demonstrates a significant correlation between the two metrics. Hence, the class-wise distance aptly reflects the difficulty of a class; a smaller value of class-wise distance tends to correlate with a lower class accuracy. Additionally, we explore the relationship between class difficulty and another potential indicator: variance. Fig. 4b illustrates the correlation between class-wise accuracy and variance. Unlike class-wise distance, there doesn't appear to be a correlation between class-wise accuracy and variance. This observation suggests that in an adversarial setting, the distance of a class plays a more crucial role in determining its difficulty than its variance. Previous studies emphasize the link between class-wise accuracy and variance, but our observations underscore the importance of class-wise distance. Therefore, we aim to address robust fairness through the lens of inter-class distance.

## 4 METHOD

In this section, based on the above analysis, we explore approaches to enhance the performance of hard classes, aiming to reduce robust accuracy disparity. Through a simple illustration, we demonstrate that, to elevate the performance of a hard class, it is more effective to shift the decision boundary between the hard class and its neighboring classes rather than between the hard class and distant classes. Subsequently, we identify an approach that effectively shifts the decision boundary for adjacent classes in adversarial training. A empirical verification confirms the efficacy of this method. Drawing upon these insights, we propose a novel methodology to improve robust fairness.

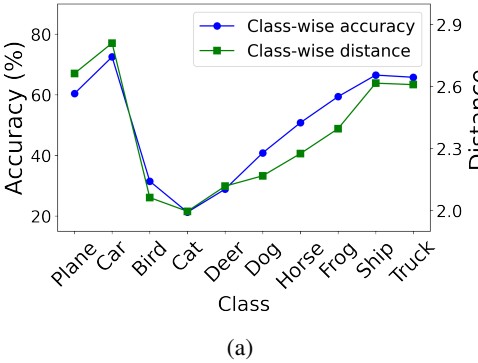 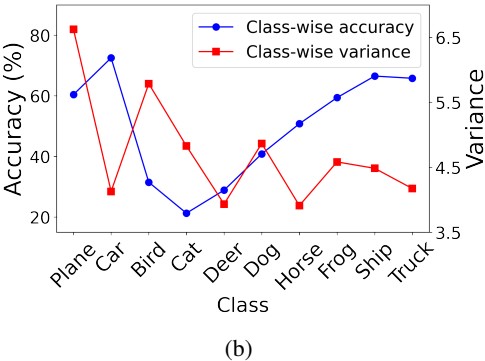

(a)                    (b)

Figure 4: (a) illustrates the test robust accuracy and the average distance from each class to the other classes in the TRADES models for CIFAR-10. (b) displays the test robust accuracy and variance for each class in the TRADES models applied to CIFAR-10 (See Appendix D.2 for further details).

## 4.1 CONSIDERATION OF CLASS-WISE DISTANCE

**Illustration example**    Our theoretical analyses in Section 3.1 demonstrate that robust fairness is exacerbated by similar classes. From this insight, we present an example illustrating that improving a hard class's performance is more effectively achieved by moving the decision boundary (DB) between the hard class and its neighboring class than that between the hard class and a distant class. Fig. 6 shows a mixture of three Gaussian distributions with identical variance. Here, we consider the distributions from left to right as classes C, A, and B respectively. Class A, positioned between classes B and C, is the hardest class. The figure displays the effect of moving the DBs, $DB_{AB}$ and $DB_{AC}$, away from class A's center. We will compare moving $DB_{AB}$ and moving $DB_{AC}$ to determine which is more efficient in enhancing the performance of class A. The shaded areas in the figure represent how much each DB needs to be shifted to achieve same performance gain for class A.

Comparatively, to achieve an equivalent performance enhancement, the shift required for $DB_{AC}$ is notably larger than for $DB_{AB}$. Moreover, considering potential performance drops in either class B or C from adjusting the DB, shifting the closer DB of class B appears more efficient for average accuracy than adjusting that of class C. This example consistently supports our intuition that an efficient strategy to mitigate accuracy disparity in a multi-classification task is to distance the DB between neighboring classes. This approach demonstrates its efficiency in terms of average accuracy, even though there might be a change in the worst class. Furthermore, in various subsequent experiments, we observe a net increase in performance.

**Our approaches**    To distance the DB from the center of the hard class A in an adversarial setting, we employ two strategies: adjusting loss weights and adversarial margins, $\epsilon$. Firstly, by assigning a greater weight to the loss of the hard class, the learning process naturally leans towards improving the performance of this class, subsequently pushing the DB away from the center of the hard class. Secondly, in terms of the adversarial margin, as it increases, the model is trained to predict a broader range of input distributions as belonging to the specified class. Allocating a larger adversarial margin to the hard class than adjacent classes pushes the DB away from the hard class's center by the margin difference. Thus, assigning a larger adversarial margin to the hard class can be leveraged as an approach to distance the DB between it and neighboring classes.

We conduct experiments with different adversarial margins across classes to verify that assigning a larger margin to a specific class, compared to its neighboring class, shifts the DB and improves the class performance. Fig. 5 shows the results of three robust classifiers trained on a binary classification task distinguishing between the bird and cat classes from CIFAR-10. Although each model has different adversarial margins during training, they are tested with a consistent $\epsilon = 6$. The findings reveal that models with identical $\epsilon$ values perform similarly. However, when trained with varying $\epsilon$ values, both the clean and robust accuracy metrics highlight superior performance for the class with the larger margin. This suggests that to boost a class's performance, it's beneficial to allocate the hard class a larger adversarial margin. Further details on Fig. 5 and theoretical analysis of the adversarial margin's impact are available in Appendices D.2 and C.3.

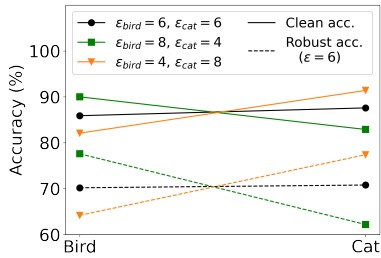

Figure 5: Results of binary classification tasks between bird and cat.

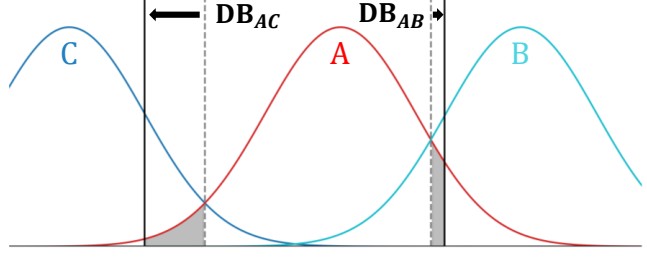

Figure 6: Illustration example of classification of three gaussian distributions with varying means. The shaded areas in the figure are of equal size.

## 4.2 DISTANCE-AWARE FAIR ADVERSARIAL TRAINING

Building on the aforementioned approaches, we introduce the Distance-Aware Fair Adversarial Training (DAFA) methodology, which incorporates class-wise distance to bolster robust fairness. We will establish a framework to assign loss weights and adversarial margins to each class based on inter-class similarities. Initially, we employ the class-wise softmax output probability as a measure of these similarities. The average class-wise softmax output probability across all class data can encapsulate inter-class distance relationships (Hinton et al., 2015). Hence, close classes tend to show higher probability values, whereas unrelated ones would have lower probability values. For brevity, we refer to the average class-wise softmax output probability as 'class-wise probability'. This class-wise probability is denoted as $p_i(j)$, where $i$ stands for the true class and $j$ for the predicted class. Here, $p_i(i)$ signifies the probability of being correctly classified, representing the difficulty of the class. Utilizing the class-wise probability, we formulate a strategy to allocate weights that prioritize the hard class in both the loss and adversarial margin. Each class weight in the dataset is denoted as $\mathcal{W}_y$, where $y \in \{1, ..., C\}$ and $C$ is the number of classes in the dataset. We devise a new weight allocation approach: we reduce the weight of the easier class by a specific amount and simultaneously increase the weight of the harder class by the same magnitude.

Specifically, considering two classes $i$ and $j$, we begin by comparing $p_i(i)$ and $p_j(j)$. Once we determine which class is relatively easier, we decide on the amount of weight to subtract from the easier class and add to the weight of the harder class. In this context, taking into account the similarity between the classes, we determine the change in weight based on the similarity values between class $i$ and $j$, i.e. $p_i(j)$ or $p_j(i)$. Here, we determine the magnitude of weight change using the hard class as a reference. That is, when class $i$ is harder than $j$, we subtract $p_i(j)$ from $\mathcal{W}_j$ and add $p_i(j)$ to $\mathcal{W}_i$. Taking into account the opposite scenario as well, the formula for computing the weight is as follows:

$$\begin{cases} \mathcal{W}_i \leftarrow \mathcal{W}_i + p_i(j) \\ \mathcal{W}_j \leftarrow \mathcal{W}_j - p_i(j) \end{cases} \text{ if } p_i(i) < p_j(j), \qquad \begin{cases} \mathcal{W}_i \leftarrow \mathcal{W}_i - p_j(i) \\ \mathcal{W}_j \leftarrow \mathcal{W}_j + p_j(i) \end{cases} \text{ if } p_i(i) > p_j(j). \quad (6)$$

Applying the above formula to a general classification task, the weight for class $i$ is computed as follows:

$$\mathcal{W}_i = \mathcal{W}_{i,0} + \sum_{j \neq i} \mathbf{1}\{p_i(i) < p_j(j)\} \cdot p_i(j) - \mathbf{1}\{p_i(i) > p_j(j)\} \cdot p_j(i) \quad (7)$$

$\mathbf{1}\{\cdot\}$ denotes an indicator function. $\mathcal{W}_{i,0}$ signifies the initial weight of class $i$. We set the initial weight for all classes to 1. Furthermore, when nearby classes are all at a similar distance from the hard class, we observe that drawing more weight from the relatively easier classes among them leads to better outcomes. Based on these observations, we determine the weights for each class as follows:

$$\mathcal{W}_i = 1 + \lambda \sum_{j \neq i} \mathbf{1}\{p_i(i) < p_j(j)\} \cdot p_i(j)p_j(j) - \mathbf{1}\{p_i(i) > p_j(j)\} \cdot p_j(i)p_i(i) \quad (8)$$

Here, $\lambda$ is the hyperparameter that controls the scale of the change in $\mathcal{W}_i$. Eq. 8 results from multiplying the class difficulty $p_j(j)$ or $p_i(i)$ with the changing weights $p_i(j)$ and $p_j(i)$. Finally, incorporating the weights calculated according to Eq. 8 with the two approaches—loss weight and adversarial margin—we propose the following adversarial training loss by rewriting Eq. 1.

$$\mathcal{L}(\boldsymbol{x}, y; \mathcal{W}_y) = \mathcal{W}_y \mathcal{L}_{CE}(f_\theta(\boldsymbol{x}), y) + \beta \max_{\|\boldsymbol{\delta}\| \leq \mathcal{W}_y \epsilon} \text{KL}\big(f_\theta(\boldsymbol{x}), f_\theta(\boldsymbol{x} + \boldsymbol{\delta})\big). \quad (9)$$

Table 1: Performance of the models for improving robust fairness methods. The best results among robust fairness methods are indicated in bold.

(a) CIFAR-10

| Method | Clean | | | Robust | | |
|---|---|---|---|---|---|---|
| | Average | Worst | $\rho_{nat}$ | Average | Worst | $\rho_{rob}$ |
| TRADES | $82.04 \pm 0.10$ | $65.70 \pm 1.32$ | $0.00$ | $49.80 \pm 0.18$ | $21.31 \pm 1.19$ | $0.00$ |
| TRADES + FRL | $82.97 \pm 0.30$ | $71.94 \pm 0.97$ | $0.11$ | $44.68 \pm 0.27$ | $25.84 \pm 1.09$ | $0.11$ |
| TRADES + CFA | $79.81 \pm 0.15$ | $65.18 \pm 0.16$ | $-0.04$ | $\mathbf{50.78 \pm 0.05}$ | $27.62 \pm 0.04$ | $0.32$ |
| TRADES + BAT | $\mathbf{87.16 \pm 0.07}$ | $\mathbf{74.02 \pm 1.38}$ | $\mathbf{0.19}$ | $45.89 \pm 0.20$ | $17.82 \pm 1.09$ | $-0.24$ |
| TRADES + WAT | $80.97 \pm 0.29$ | $70.57 \pm 1.54$ | $0.06$ | $46.30 \pm 0.27$ | $28.55 \pm 1.30$ | $0.27$ |
| TRADES + DAFA | $81.63 \pm 0.20$ | $67.90 \pm 1.25$ | $0.03$ | $49.05 \pm 0.14$ | $\mathbf{30.53 \pm 1.04}$ | $\mathbf{0.42}$ |

(b) CIFAR-100

| Method | Clean | | | Robust | | |
|---|---|---|---|---|---|---|
| | Average | Worst | $\rho_{nat}$ | Average | Worst | $\rho_{rob}$ |
| TRADES | $58.34 \pm 0.38$ | $16.80 \pm 1.05$ | $0.00$ | $25.60 \pm 0.14$ | $1.33 \pm 0.94$ | $0.00$ |
| TRADES + FRL | $57.59 \pm 0.34$ | $18.73 \pm 1.84$ | $0.10$ | $24.93 \pm 0.29$ | $1.80 \pm 0.54$ | $0.33$ |
| TRADES + CFA | $57.74 \pm 1.31$ | $15.00 \pm 1.55$ | $-0.12$ | $23.49 \pm 0.30$ | $1.67 \pm 1.25$ | $0.17$ |
| TRADES + BAT | $\mathbf{62.80 \pm 0.17}$ | $\mathbf{22.40 \pm 2.15}$ | $\mathbf{0.41}$ | $23.42 \pm 0.08$ | $0.20 \pm 0.40$ | $-0.93$ |
| TRADES + WAT | $53.56 \pm 0.43$ | $17.07 \pm 2.29$ | $-0.07$ | $22.65 \pm 0.28$ | $1.87 \pm 0.19$ | $0.29$ |
| TRADES + DAFA | $58.07 \pm 0.05$ | $18.67 \pm 0.47$ | $0.11$ | $\mathbf{25.08 \pm 0.10}$ | $\mathbf{2.33 \pm 0.47}$ | $\mathbf{0.73}$ |

(c) STL-10

| Method | Clean | | | Robust | | |
|---|---|---|---|---|---|---|
| | Average | Worst | $\rho_{nat}$ | Average | Worst | $\rho_{rob}$ |
| TRADES | $61.13 \pm 0.57$ | $39.29 \pm 1.71$ | $0.00$ | $31.36 \pm 0.51$ | $7.73 \pm 0.99$ | $0.00$ |
| TRADES + FRL | $56.75 \pm 0.37$ | $31.26 \pm 1.41$ | $-0.28$ | $28.99 \pm 0.39$ | $7.94 \pm 0.49$ | $-0.05$ |
| TRADES + CFA | $\mathbf{60.70 \pm 0.57}$ | $38.53 \pm 0.95$ | $-0.03$ | $\mathbf{31.88 \pm 0.14}$ | $7.69 \pm 0.27$ | $0.01$ |
| TRADES + BAT | $59.06 \pm 1.29$ | $36.75 \pm 2.44$ | $-0.10$ | $23.37 \pm 0.98$ | $3.22 \pm 1.00$ | $-0.84$ |
| TRADES + WAT | $52.92 \pm 1.80$ | $30.82 \pm 1.44$ | $-0.35$ | $26.98 \pm 0.25$ | $6.72 \pm 0.75$ | $-0.27$ |
| TRADES + DAFA | $60.50 \pm 0.57$ | $\mathbf{42.23 \pm 2.26}$ | $\mathbf{0.06}$ | $29.98 \pm 0.46$ | $\mathbf{10.73 \pm 1.31}$ | $\mathbf{0.34}$ |

In Eq. 9, the weights we compute are multiplied with both the cross-entropy loss and the adversarial margin. Although we could multiply the weights to the KL divergence term to promote enhanced robustness for the worst class, a previous study (Wei et al., 2023) indicates that using a large $\beta$ for the worst class can impair average clean accuracy without improving its robustness—we observe results that align with this finding. During the process of DAFA, similar to earlier methods (Xu et al., 2021; Wei et al., 2023), we apply our approach after an initial warm-up phase. Detailed algorithms of our methodology can be found in Appendix A.

## 5 EXPERIMENT

**Implementation details**  We conducted experiments on CIFAR-10, CIFAR-100 (Krizhevsky et al., 2009), and STL-10 (Coates et al., 2011) using TRADES (Zhang et al., 2019) as our baseline adversarial training algorithm and ResNet18 (He et al., 2016b) for the model architecture. For our method, the warm-up epoch was set to $\tau = 70$ and the hyperparameter $\lambda$ was set to $\lambda = 1$ for CIFAR-10 and $\lambda = 1.5$ for CIFAR-100 and STL-10 due to the notably low performance of hard classes. Robustness was evaluated using the adaptive auto attack ($A^3$) (Liu et al., 2022). Since class-wise accuracy tends to vary more than average accuracy, we ran each test three times with distinct seeds and showcased the mean accuracy from the final five epochs—effectively averaging over 15 models. To ensure a fair comparison in our comparative experiments with existing methods, we employed the

original code from each approach. More specifics can be found in Appendix D.1. Additionally, the results of experiments comparing each method both in each method's setting and the same setting are provided in the Appendix F.

**Improvement in robust fairness performance**   We compared the TRADES baseline model with five different robust fairness improvement methodologies: FRL (Xu et al., 2021), CFA (Wei et al., 2023), BAT (Sun et al., 2023), and WAT (Li & Liu, 2023), as well as our own approach. Both average and worst performances for clean accuracy and robust accuracy were measured. Additionally, to assess the change in worst accuracy performance against the change in average accuracy performance of the baseline model, we adopted the evaluation metric $\rho$ proposed by Li & Liu (2023) (refer to Appendix D.1). A higher value of $\rho$ indicates superior method performance.

As summarized in Table 1a for the CIFAR-10 dataset, most robust fairness improvement methodologies demonstrate an enhancement in the worst class's robust accuracy when compared to the baseline model. Yet, the existing methods reveal a trade-off: a significant increase in the worst class's robust accuracy often comes at the cost of a noticeable decline in the average robust accuracy, and vice versa. In contrast, our methodology improves the worst class's robust accuracy by over 9%p while maintaining the average robust accuracy close to that of the baseline model. Additionally, we conducted experiments incorporating exponential moving average (EMA) Izmailov et al. (2018) into our method, following the approach of the previous method (CFA) that utilized a variant of EMA. As shown in Table 11b, when EMA is employed, our method demonstrates robust average accuracy comparable to the existing method while significantly increasing robust worst accuracy.

From Tables 1b and 1c, it's evident that for datasets where worst robust accuracy is notably low, most existing methodologies struggle to enhance it. However, similar to the results on the CIFAR-10 dataset, our approach successfully elevates the worst robust accuracy while preserving the average robust accuracy. Moreover, in terms of clean accuracy, conventional methods lead to a marked decrease in average clean accuracy. In contrast, our approach preserves a clean accuracy performance comparable to the baseline. These results underscore the efficiency of our approach in significantly enhancing the performance of the worst class without detrimentally affecting the performance of other classes compared to conventional methods. By comparing the $\rho_{rob}$ results across all three datasets, we can confirm that our method considerably improves robust fairness compared to existing techniques.

**Effectiveness of DAFA**   Through DAFA, we conducted experiments to ascertain whether the two class-wise parameters applied to the loss $\mathcal{L}_{CE}$ and adversarial margin $\epsilon_y$ effectively improve robust fairness. Table 2 presents the results of individually applying each class-wise parameter as well as applying both concurrently. The results indicate that both class-wise parameters are effective in enhancing the worst class accuracy. When combined, the improvement is roughly equal to the sum of the enhancements obtained from each parameter individually. Ad-

Table 2: Performance comparison of our methods when applied only to the loss and when applied only to the adversarial margin.

| Method | Robust | |
| --- | --- | --- |
| | Average | Worst |
| TRADES | $49.80 \pm 0.18$ | $21.31 \pm 1.19$ |
| DAFA$_{\mathcal{L}_{CE}}$ | $49.41 \pm 0.15$ | $28.10 \pm 1.26$ |
| DAFA$_\epsilon$ | $49.51 \pm 0.15$ | $23.64 \pm 0.85$ |
| DAFA | $49.05 \pm 0.14$ | $30.53 \pm 1.04$ |

ditionally, we have verified that DAFA performs well when applied to standard adversarial training (PGD) (Madry et al., 2017) or other architectures. Results and analyses of various ablation studies, including the warm-up epoch and the hyperparameter $\lambda$, can be found in Appendix E.

## 6   CONCLUSION

In this study, we analyze robust fairness from the perspective of inter-class similarity. We conduct both theoretical and empirical analyses using class distance, uncovering the interrelationship between class-wise accuracy and class distance. We demonstrate that to enhance the performance of the hard class, classes that are closer in distance should be given more consideration than distant classes. Building upon these findings, we introduce Distance Aware Fair Adversarial training (DAFA), which considers inter-class similarities to improve robust fairness. Our methodology displays superior average robust accuracy and worst robust accuracy performance across various datasets and model architectures when compared to existing methods.

## 7 ACKNOWLEDGEMENT

This work was supported by the National Research Foundation of Korea (NRF) grants funded by the Korea government (Ministry of Science and ICT, MSIT) (2022R1A3B1077720 and 2022R1A5A708390811), Institute of Information & Communications Technology Planning & Evaluation (IITP) grants funded by the Korea government (MSIT) (2021-0-01343: Artificial Intelligence Graduate School Program (Seoul National University) and 2022-0-00959), AI Convergence Innovation Human Resources Development (EWU) (RS-2022-00155966), and the BK21 FOUR program of the Education and Research Program for Future ICT Pioneers, Seoul National University in 2023.

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

## A  THE ALGORITHMS OF DAFA

---

**Algorithm 1** Training procedure of DAFA

---

**Require:** Dataset $\mathcal{D}$, model parameter $\boldsymbol{\theta}$, batch size $n$, warm-up epoch $\tau$, training epoch $K$, learning rate $\alpha$, class-wise probability $p$
1: nominal adversarial training:
2: **for** $t = 1$ **to** $\tau$ **do**
3:    $\{\boldsymbol{x}_i, y_i\}_{i=1}^n \sim D$
4:    $\theta \leftarrow \arg\min_\theta \mathcal{L}(\boldsymbol{x}_i + \boldsymbol{\delta}_i, y_i)$
5:    $p \leftarrow f(\boldsymbol{x}_i + \boldsymbol{\delta}_i, y_i)$
6: **end for**
7: compute parameter weights:
8: $\mathcal{W} \leftarrow \text{DAFA}_{comp}(p)$
9: **for** $t = \tau + 1$ **to** $K$ **do**
10:    $\{\boldsymbol{x}_i, y_i\}_{i=1}^n \sim D$
11:    $\theta \leftarrow \arg\min_\theta \mathcal{L}(\boldsymbol{x}_i + \boldsymbol{\delta}_i, y_i; \mathcal{W}_{y_i})$
12: **end for**
13: return $f(; \theta)$

---

**Algorithm 2** DAFA$_{comp}$

---

**Require:** Number of class $C$, class-wise probability $p$, Scale parameter $\lambda$
1: $\mathcal{W} \leftarrow \mathbf{1}$
2: **for** $i = 1$ **to** $C$ **do**
3:    **for** $j = 1$ **to** $C$ **do**
4:       **if** $i == j$ **then**
5:          continue
6:       **else if** $p_i(i) < p_j(j)$ **then**
7:          $\mathcal{W}_i \leftarrow \mathcal{W}_i + \lambda p_i(j) \cdot p_j(j)$
8:       **else if** $p_i(i) > p_j(j)$ **then**
9:          $\mathcal{W}_i \leftarrow \mathcal{W}_i - \lambda p_j(i) \cdot p_i(i)$
10:      **end if**
11:    **end for**
12: **end for**
13: return $\mathcal{W}$

---

During model training via adversarial training, the method is applied post a warm-up training phase. Additionally, since we determine the weight values using the softmax output probability results of adversarial examples computed during training, there is no additional inference required for the computation of these weight values. In this context, to determine the weight value for each class using Eq. 8 at every epoch, a computational cost of $\mathcal{O}(C^2)$ is incurred, where $C$ represents the number of classes. Recognizing this, we compute the training parameters solely once, following the warm-up training. As a result, our methodology can be integrated with minimal additional overhead.

## B  RELATED WORK

**Adversarial robustness**  Adversarial training uses adversarial examples to train the model robust to adversarial examples (Madry et al., 2017). For a dataset $\mathcal{D} = \{(\boldsymbol{x}_i, y_i)\}_{i=1}^n$, where $\boldsymbol{x}_i \in \mathbb{R}^d$ is clean data and each $y_i$ from the class set indicates its label, the equation of adversarial training can be expressed as the subsequent optimization framework:

$$\min_\theta \sum_{i=1}^n \max_{\|\boldsymbol{\delta}_i\| \le \epsilon} \mathcal{L}_{CE}(f_\theta(\boldsymbol{x}_i + \boldsymbol{\delta}_i), y_i). \tag{10}$$

Here, $\theta$ denotes the parameters of the model $f$, and the loss function is represented by $\mathcal{L}$. Additionally, $\epsilon$ is termed the adversarial margin. This margin sets the maximum allowable limit for adversarial perturbations. An alternative approach to adversarial training is the TRADES method (Zhang et al., 2019). In addition, various methodologies have been proposed to defend against adversarial attacks, encompassing approaches that leverage different adversarial training algorithms (Wang et al., 2019; Wu et al., 2020; Zhang et al., 2020), methodologies that enhance robustness performance through the utilization of additional data (Lee et al., 2021; Carmon et al., 2019; Lee & Lee, 2023; Gowal et al., 2021), and methodologies employing data augmentation (Rebuffi et al., 2021; Li & Spratling, 2023).

For the attacks, as mentioned in the main paper, the fast gradient sign method (FGSM) stands as a prevalent technique for producing adversarial samples (Goodfellow et al., 2014). PGD can be seen as an extended iteration of FGSM (Madry et al., 2017). AutoAttack (Croce & Hein, 2020) and adaptive auto attack ($\text{A}^3$) (Liu et al., 2022) are an efficient and reliable attacks. We employed $\text{A}^3$ to assess the robust accuracy of various models.

**Robust fairness**  Benz et al. (2021) posited that the issue of robust fairness resembles the accuracy disparity problem arising from class imbalances when training on LT distribution datasets. Based on this premise, they applied solutions designed for handling LT distributions to address robust fairness. As a result, while the performance of relatively easier classes decreased, the performance of more challenging classes improved. Tian et al. (2021) highlighted the robust fairness issue and, through experiments across various datasets and algorithms, observed that this issue commonly manifests in adversarial settings. Xu et al. (2021) perceived the robust fairness issue as emerging from differences in class-wise difficulty and theoretically analyzed it by equating it to variance differences in class-wise distribution. Building on this analysis, they introduced a regularization methodology proportional to class-wise training performance, enhancing the accuracy of the worst-performing class. Ma et al. (2022) revealed a trade-off between robust fairness and robust average accuracy, noting that this trade-off intensifies as the training adversarial margin $\epsilon$ increases.

Wei et al. (2023) identified that datasets have class-specific optimal regularization parameters or $\epsilon$ values for adversarial training. Utilizing both training and validation performances, they proposed a method that adaptively allocates appropriate regularization to each class during the learning process, thereby improving robust fairness. Hu et al. (2023) conducted a theoretical examination of accuracy disparity in robustness from a data imbalance perspective and, through experiments on various datasets, demonstrated the relationship between data imbalance ratio and robust accuracy disparity. Li & Liu (2023) suggested a methodology that uses a validation set to evaluate the model and update the loss weights for each class, subsequently enhancing the performance of the worst-performing class. Sun et al. (2023) divided robust fairness into source class fairness and target class fairness. By proposing a training method that balances both, they successfully reduced the performance disparities between classes.

## C  THEORETICAL ANALYSIS

### C.1  PROOF OF THE THEOREMS

In this section, we provide proof for the theorems and corollaries presented in our main paper. We begin by detailing the proof for the newly defined distribution, $\mathcal{D}'$:

$$y \overset{u.a.r}{\sim} \{-1, +1\}, \ \boldsymbol{\mu} = (\eta, \eta, \cdots, \eta), \qquad \boldsymbol{x} \sim \begin{cases} \mathcal{N}(\gamma\boldsymbol{\mu}, \sigma^2 I), & \text{if } y = +1 \\ \mathcal{N}(-\gamma\boldsymbol{\mu}, I), & \text{if } y = -1 \end{cases}. \tag{11}$$

Here, $I$ denotes an identity matrix with a dimension of $d$, and $\sigma^2$ represents the variance of the distribution for the class $y = +1$, where $\sigma^2 > 1$. $\gamma$ is constrained to be greater than zero. We initiate with a lemma to determine the weight vector of an optimal linear model for binary classification under the data distribution $\mathcal{D}'$.

**Lemma 1.** *Given the data distribution $\mathcal{D}'$ as defined in equation 11, if an optimal classifier $f$ minimizes the standard error for binary classification, then the classifier will have the optimal weight vector $\boldsymbol{w} = (\frac{1}{\sqrt{d}}, \frac{1}{\sqrt{d}}, ..., \frac{1}{\sqrt{d}})$.*

For the classification task discussed in Section 3.1, the weight of the classifier adheres to the condition $\|\boldsymbol{w}\|_2 = 1$. We establish Lemma 1 following the approach used in Xu et al. (2021); Tsipras et al. (2018).

*Proof.* Given a weight vector $\boldsymbol{w} = (w_1, w_2, ..., w_d)$, we aim to prove that the optimal weight satisfies $w_1 = w_2 = \cdots = w_d$. Let's assume, for the sake of contradiction, that the optimal weight of classifier $f$ doesn't meet the condition $w_1 = w_2 = \cdots = w_d$. Hence, for some $i \neq j$ and $i, j \in \{1, 2, ..., d\}$, let's say $w_i < w_j$. This implies that the optimal classifier $f$ has a particular standard error corresponding to the optimal weight $\boldsymbol{w}$.

$$\mathcal{R}_{nat}(f|+1) + \mathcal{R}_{nat}(f|-1) = Pr.\big(\sum_{k \neq i, k \neq j}^{d} w_k \mathcal{N}(\gamma\eta, \sigma^2) + b + w_i \mathcal{N}(\gamma\eta, \sigma^2) + w_j \mathcal{N}(\gamma\eta, \sigma^2) < 0\big)$$

$$+ Pr.\big(\sum_{k \neq i, k \neq j}^{d} w_k \mathcal{N}(-\gamma\eta, 1) + b + w_i \mathcal{N}(-\gamma\eta, 1) + w_j \mathcal{N}(-\gamma\eta, 1) > 0\big). \quad (12)$$

Now, consider a new classifier $f'$, which has the same weight vector as classifier $f$ but with $w_i$ replaced by $w_j$. The resulting standard error for this new classifier $f'$ can then be described as follows:

$$\mathcal{R}_{nat}(f|+1) + \mathcal{R}_{nat}(f|-1) = Pr.\big(\sum_{k \neq i, k \neq j}^{d} w_k \mathcal{N}(\gamma\eta, \sigma^2) + b + w_j \mathcal{N}(\gamma\eta, \sigma^2) + w_j \mathcal{N}(\gamma\eta, \sigma^2) < 0\big)$$

$$+ Pr.\big(\sum_{k \neq i, k \neq j}^{d} w_k \mathcal{N}(-\gamma\eta, 1) + b + w_j \mathcal{N}(-\gamma\eta, 1) + w_j \mathcal{N}(-\gamma\eta, 1) > 0\big). \quad (13)$$

Given that $Pr.(w_i \mathcal{N}(\gamma\eta, 1) > 0) < Pr.(w_j \mathcal{N}(\gamma\eta, 1) > 0)$, the standard error for each class under the new classifier $f'$ is lower than that of $f$. This contradicts our initial assumption that the weight of the classifier $f$ is optimal. Therefore, the optimal weight must satisfy $w_1 = w_2 = \cdots = w_d$. Since we've constrained $\|\boldsymbol{w}\|_2 = 1$, it follows that $\boldsymbol{w} = (\frac{1}{\sqrt{d}}, \frac{1}{\sqrt{d}}, ..., \frac{1}{\sqrt{d}})$. $\square$

Next, we will prove the following lemma.

**Lemma 2.** *The optimal standard error for the binary classification task in Lemma 1 is identical to the optimal standard error for the binary classification task of the distribution obtained by translating $\mathcal{D}'$.*

*Proof.* Let $\zeta$ represent the magnitude of the translation. Then, the standard error for binary classification of the translated distribution is as follows:

$$\mathcal{R}_{nat}(g|+1) + \mathcal{R}_{nat}(g|-1) = Pr.\big(\sum_{k}^{d} w_k' \mathcal{N}(\gamma\eta + \zeta, \sigma^2) + b' < 0\big)$$

$$+ Pr.\big(\sum_{k}^{d} w_k' \mathcal{N}(-\gamma\eta + \zeta, 1) + b' > 0\big). \quad (14)$$

Let $\boldsymbol{w}' = (w_1', w_2', ..., w_d')$ represent the weight parameters and let $b'$ be the bias parameter of the classifier $g$ for the given classification task. Assuming that $\|\boldsymbol{w}'\|_2 = 1$ as previously mentioned, the equation can be reformulated as:

$$\mathcal{R}_{nat}(g|+1) + \mathcal{R}_{nat}(g|-1) = Pr.\big(\sum_k^d w_k' \mathcal{N}(\gamma\eta, \sigma^2) + \sum_k^d w_k'\zeta + b' < 0\big)$$

$$+ Pr.\big(\sum_k^d w_k' \mathcal{N}(-\gamma\eta, 1) + \sum_k^d w_k'\zeta + b' > 0\big). \quad (15)$$

Given that the equation above matches the standard error of the binary classification for the data distribution $\mathcal{D}'$ when $b'$ is replaced as $b' = b - \sum_k^d w_k'\zeta$, it can be deduced that the optimal standard error for binary classification of the translated distribution of $\mathcal{D}'$ is identical to that for $\mathcal{D}'$. $\square$

To establish Theorem 1, we leverage the aforementioned lemmas.

**Theorem 1.** *Let $0 < \epsilon < \eta$ and $A \equiv \frac{2\sqrt{d}\sigma}{\sigma^2 - 1} > 0$. Given a data distribution $\mathcal{D}$ as described in equation 2, $f_{nat}$ and $f_{rob}$ exhibit the following standard and robust errors, respectively:*

$$\mathcal{R}_{nat}(f_{nat}|+1) = Pr.\left(\mathcal{N}(0,1) < -A(\frac{1+\alpha}{2}\eta) + \sqrt{(\frac{A}{\sigma})^2(\frac{1+\alpha}{2}\eta)^2 + \frac{2\log\sigma}{\sigma^2 - 1}}\right), \quad (16)$$

$$\mathcal{R}_{rob}(f_{rob}|+1) = Pr.\left(\mathcal{N}(0,1) < -A(\frac{1+\alpha}{2}\eta - \epsilon) + \sqrt{(\frac{A}{\sigma})^2(\frac{1+\alpha}{2}\eta - \epsilon)^2 + \frac{2\log\sigma}{\sigma^2 - 1}}\right). \quad (17)$$

*Consequently, both $\mathcal{R}_{nat}(f_{nat}|+1)$ and $\mathcal{R}_{rob}(f_{rob}|+1)$ decrease monotonically with respect to $\alpha$.*

*Proof.* In light of Lemma 2, the distribution $\mathcal{D}$ for binary classification can be translated by an amount of $\frac{\alpha-1}{2}\eta$ as follows:

$$y \overset{u.a.r}{\sim} \{-1, +1\}, \quad \boldsymbol{\mu} = (\eta, \eta, \cdots, \eta), \quad \boldsymbol{x} \sim \begin{cases} \mathcal{N}(\frac{1+\alpha}{2}\boldsymbol{\mu}, \sigma^2 I), & \text{if } y = +1 \\ \mathcal{N}(-\frac{1+\alpha}{2}\boldsymbol{\mu}, 1), & \text{if } y = -1 \end{cases}. \quad (18)$$

Based on Lemma 1, the optimal classifier $f_{nat}$ for standard error has the optimal parameters, with $\boldsymbol{w} = (\frac{1}{\sqrt{d}}, \frac{1}{\sqrt{d}}, ..., \frac{1}{\sqrt{d}})$. Consequently, the standard errors of $f_{nat}$ for $y = +1$ and $y = -1$ can be expressed as:

$$\mathcal{R}_{nat}(f_{nat}|+1) = Pr.\left(\sum_{i=1}^d w_i \mathcal{N}(\frac{1+\alpha}{2}\eta, \sigma^2) + b < 0\right) = Pr.\left(\mathcal{N}(0,1) < -\frac{\sqrt{d}}{\sigma}(\frac{1+\alpha}{2}\eta) - \frac{b}{\sigma}\right), \quad (19)$$

$$\mathcal{R}_{nat}(f_{nat}|-1) = Pr.\left(\sum_{i=1}^d w_i \mathcal{N}(-\frac{1+\alpha}{2}\eta, 1) + b > 0\right) = Pr.\left(\mathcal{N}(0,1) < -\sqrt{d}(\frac{1+\alpha}{2}\eta) + b\right). \quad (20)$$

We aim to minimize the combined error of the two aforementioned terms to optimize the classifier $f_{nat}$ with respect to the parameter $b$. To do this, we follow the approach used in (Xu et al., 2021), which identifies the point where $\frac{\partial}{\partial b}\big(\mathcal{R}_{nat}(f_{nat}|+1) + \mathcal{R}_{nat}(f_{nat}|-1)\big) = 0$. Given that the cumulative distribution function of the normal distribution is expressed as $Pr.(\mathcal{N}(0,1) < z) = \int_{-\infty}^z \exp(-\frac{t^2}{2})dt$, the derivative of the combined error concerning $b$ is as follows:

$$\rightarrow \exp\left(-\frac{1}{2}(-\frac{\sqrt{d}}{\sigma}(\frac{1+\alpha}{2}\eta) - \frac{b}{\sigma})^2\right)(-\frac{1}{\sigma}) + \exp\left((-\frac{1}{2}(-\sqrt{d}(\frac{1+\alpha}{2}\eta) + b)^2\right) = 0, \quad (21)$$

$$\overset{\times -\sigma}{\rightarrow} \sigma\exp\left((-\frac{1}{2}(-\sqrt{d}(\frac{1+\alpha}{2}\eta) + b)^2\right) = \exp\left(-\frac{1}{2}(-\frac{\sqrt{d}}{\sigma}(\frac{1+\alpha}{2}\eta) - \frac{b}{\sigma})^2\right), \quad (22)$$

$$\overset{\log}{\rightarrow} \log\sigma - \frac{1}{2}(-\sqrt{d}(\frac{1+\alpha}{2}\eta) + b)^2 = -\frac{1}{2}(-\frac{\sqrt{d}}{\sigma}(\frac{1+\alpha}{2}\eta) - \frac{b}{\sigma})^2. \quad (23)$$

Then, the point where $\frac{\partial}{\partial b}\big(\mathcal{R}_{nat}(f_{nat}|+1) + \mathcal{R}_{nat}(f_{nat}|-1)\big) = 0$ is given as

$$b = \frac{\sigma^2 + 1}{\sigma^2 - 1} \cdot \sqrt{d}(\frac{1+\alpha}{2}\eta) - \sigma\sqrt{\frac{4}{(\sigma^2-1)^2} \cdot d(\frac{1+\alpha}{2}\eta)^2 + \frac{2\log\sigma}{\sigma^2-1}}. \tag{24}$$

By employing the optimal parameter $b$, the standard error of the optimal classifier $f_{nat}$ can be reformulated.

$$\mathcal{R}_{nat}(f_{nat}|+1) = Pr.\left(\mathcal{N}(0,1) < -A(\frac{1+\alpha}{2}\eta) + \sqrt{(\frac{A}{\sigma})^2(\frac{1+\alpha}{2}\eta)^2 + \frac{2\log\sigma}{\sigma^2-1}}\right). \tag{25}$$

Here, $A$ is defined as $A = \frac{2\sqrt{d}\sigma}{\sigma^2-1}$. Our objective is then to demonstrate our Theorem 1: $\mathcal{R}_{nat}(f_{nat}|+1)$ monotonically decreases with $\alpha$. Leveraging the property $\sigma > 1$, it follows that $\frac{2\log\sigma}{\sigma^2-1} > 0$ and $A > 0$. The derivative of Eq. 25 with respect to $\alpha$ is:

$$\frac{\partial}{\partial\alpha}\mathcal{R}_{nat}(f_{nat}|+1) = \exp(-\frac{z_{+1}^2}{2})\left(-\frac{A}{2}\eta + \frac{\left(\frac{A}{\sigma}\right)^2\left(\frac{1+\alpha}{2}\eta\right)\frac{\eta}{2}}{\sqrt{\left(\frac{A}{\sigma}\right)^2\left(\frac{1+\alpha}{2}\eta\right)^2 + \frac{2\log\sigma}{\sigma^2-1}}}\right) \tag{26}$$

$$\leq \exp(-\frac{z_{+1}^2}{2})\left(-\frac{A}{2}\eta + \frac{\left(\frac{A}{\sigma}\right)^2\left(\frac{1+\alpha}{2}\eta\right)\frac{\eta}{2}}{\sqrt{\left(\frac{A}{\sigma}\right)^2\left(\frac{1+\alpha}{2}\eta\right)^2}}\right) \tag{27}$$

$$= \exp(-\frac{z_{+1}^2}{2})\left(-\frac{A}{2}\eta(1 - \frac{1}{\sigma})\right) < 0. \tag{28}$$

Here, $z_{+1} = -A(\frac{1+\alpha}{2}\eta) + \sqrt{\left(\frac{A}{\sigma}\right)^2\left(\frac{1+\alpha}{2}\eta\right)^2 + \frac{2\log\sigma}{\sigma^2-1}}$. Given that the derivative of $\mathcal{R}_{nat}(f_{nat}|+1)$ with respect to $\alpha$ is less than zero, $\mathcal{R}_{nat}(f_{nat}|+1)$ monotonically decreases as $\alpha$ increases.

Next, we aim to demonstrate that $\mathcal{R}_{rob}(f_{rob}|+1)$ monotonically decreases with respect to $\alpha$. Following the proof approach for $\mathcal{R}_{nat}(f_{nat}|+1)$ and referencing Lemma 2, the distribution $\mathcal{D}$ for binary classification can be shifted by an amount of $\frac{\alpha-1}{2}\eta$ as follows:

$$y \overset{u.a.r}{\sim} \{-1,+1\}, \; \boldsymbol{\mu} = (\eta, \eta, \cdots, \eta), \qquad \boldsymbol{x} \sim \begin{cases} \mathcal{N}(\frac{1+\alpha}{2}\boldsymbol{\mu} - \epsilon, \sigma^2 I), & \text{if } y = +1 \\ \mathcal{N}(-\frac{1+\alpha}{2}\boldsymbol{\mu} + \epsilon, 1), & \text{if } y = -1 \end{cases}. \tag{29}$$

Here, $\epsilon$ satisfies $0 < \epsilon < \eta$. Given that the classification task can be aligned with the task outlined in Eq. 11 by adjusting the mean of the distributions to $\frac{1+\alpha}{2}\boldsymbol{\mu} - \epsilon$, the optimal parameter $\boldsymbol{w}_{rob}$ is chosen as $\boldsymbol{w}_{rob} = (\frac{1}{\sqrt{d}}, \frac{1}{\sqrt{d}}, ..., \frac{1}{\sqrt{d}})$ and the value of $b_{rob}$ is determined as follows:

$$b_{rob} = \frac{\sigma^2 + 1}{\sigma^2 - 1} \cdot \sqrt{d}(\frac{1+\alpha}{2}\eta - \epsilon) - \sigma\sqrt{\frac{4}{(\sigma^2-1)^2} \cdot d(\frac{1+\alpha}{2}\eta - \epsilon)^2 + \frac{2\log\sigma}{\sigma^2-1}}. \tag{30}$$

Using the optimal parameter $\boldsymbol{w}_{rob}$ and $b_{rob}$, the robust error of the optimal classifier $f_{rob}$ for class $y = +1$ is expressed as follows:

$$\mathcal{R}_{rob}(f_{rob}|+1) = Pr.\left(\mathcal{N}(0,1) < -A(\frac{1+\alpha}{2}\eta - \epsilon) + \sqrt{(\frac{A}{\sigma})^2(\frac{1+\alpha}{2}\eta - \epsilon)^2 + \frac{2\log\sigma}{\sigma^2-1}}\right). \tag{31}$$

Since the derivative of Eq. 31 matches that of Eq. 25, we deduce that $\frac{\partial}{\partial\alpha}\big(\mathcal{R}_{rob}(f_{rob}|+1)\big) < 0$. Consequently, $\mathcal{R}_{rob}(f_{rob}|+1)$ decreases monotonically with respect to $\alpha$.

$\square$

We subsequently prove Theorem 2 by leveraging the error computed in Theorem 1.

**Theorem 2.** *Let $D\big(\mathcal{R}(f)\big)$ be the disparity in errors for the two classes within dataset $\mathcal{D}$ as evaluated by a classifier $f$, Then $D\big(\mathcal{R}(f)\big)$ is*

$$D\big(\mathcal{R}(f)\big) \equiv \mathcal{R}(f|+1) - \mathcal{R}(f|-1)$$

$$= Pr.\left(\frac{A}{\sigma}g(\alpha) - \sqrt{A^2 g^2(\alpha) + h(\sigma)} < \mathcal{N}(0,1) < -Ag(\alpha) + \sqrt{\left(\frac{A}{\sigma}\right)^2 g^2(\alpha) + \frac{h(\sigma)}{\sigma^2}}\right), \quad (32)$$

*where $g(\alpha) = \frac{1+\alpha}{2}\eta - \epsilon$, $A = \frac{2\sqrt{d}\sigma}{\sigma^2-1}$, and $h(\sigma) = \frac{2\sigma^2 \log \sigma}{\sigma^2-1}$. Then, the disparity between these two errors decreases monotonically with the increase of $\alpha$.*

*Proof.* Based on the proof of Theorem 1, the robust error of the robust model for class $y = -1$ is given as:

$$\mathcal{R}_{rob}(f_{rob}|-1) = Pr.\left(\mathcal{N}(0,1) < \frac{A}{\sigma}(\frac{1+\alpha}{2}\eta - \epsilon) - \sqrt{A^2\big(\frac{1+\alpha}{2}\eta - \epsilon\big)^2 + \frac{2\sigma^2 \log \sigma}{\sigma^2 - 1}}\right). \quad (33)$$

Given that $\frac{2\sigma^2 \log \sigma}{\sigma^2-1} > 0$, the robust error for class $y = +1$ exceeds that for class $y = -1$, implying $\mathcal{R}_{rob}(f_{rob}|+1) > \mathcal{R}_{rob}(f_{rob}|-1)$. Consequently, the discrepancy in robust errors between the classes for the robust model can be formulated as:

$$\mathcal{R}_{rob}(f_{rob}|+1) - \mathcal{R}_{rob}(f_{rob}|-1)$$

$$= Pr.\left(\frac{A}{\sigma}g(\alpha) - \sqrt{A^2 g^2(\alpha) + h(\sigma)} < \mathcal{N}(0,1) < -Ag(\alpha) + \sqrt{\left(\frac{A}{\sigma}\right)^2 g^2(\alpha) + \frac{h(\sigma)}{\sigma^2}}\right), \quad (34)$$

where $g(\alpha) = \frac{1+\alpha}{2}\eta - \epsilon$, $A = \frac{2\sqrt{d}\sigma}{\sigma^2-1}$, and $h(\sigma) = \frac{2\sigma^2 \log \sigma}{\sigma^2-1}$. We aim to demonstrate that the derivative of the aforementioned function with respect to $\alpha$ is less than zero. This implies that the discrepancy between class errors decreases monotonically with respect to $\alpha$. Then, the derivative, $\frac{\partial}{\partial \alpha}(\mathcal{R}_{rob}(f_{rob}|+1) - \mathcal{R}_{rob}(f_{rob}|-1))$, can be expressed as:

$$\exp(-\frac{z_{+1}^2}{2})\left(-A \cdot \frac{\eta}{2} + \frac{\left(\frac{A}{\sigma}\right)^2 g(\alpha)\frac{\eta}{2}}{\sqrt{\left(\frac{A}{\sigma}\right)^2 g^2(\alpha) + \frac{h(\sigma)}{\sigma^2}}}\right) - \exp(-\frac{z_{-1}^2}{2})\left(\frac{A}{\sigma} \cdot \frac{\eta}{2} - \frac{A^2 g(\alpha)\frac{\eta}{2}}{\sqrt{A^2 g^2(\alpha) + h(\sigma)}}\right)$$

$$< \exp(-\frac{z_{-1}^2}{2})\left(-A \cdot \frac{\eta}{2} + \frac{\left(\frac{A}{\sigma}\right)^2 g(\alpha)\frac{\eta}{2}}{\sqrt{\left(\frac{A}{\sigma}\right)^2 g^2(\alpha) + \frac{h(\sigma)}{\sigma^2}}} - \frac{A}{\sigma} \cdot \frac{\eta}{2} + \frac{A^2 g(\alpha)\frac{\eta}{2}}{\sqrt{A^2 g^2(\alpha) + h(\sigma)}}\right)$$

$$= \exp(-\frac{z_{-1}^2}{2})\left(-A \cdot \frac{\eta}{2}\right)\left((1 + \frac{1}{\sigma}) - (\frac{\frac{A}{\sigma}g(\alpha)}{\sqrt{A^2 g^2(\alpha) + h(\sigma)}} + \frac{Ag(\alpha)}{\sqrt{A^2 g^2(\alpha) + h(\sigma)}})\right)$$

$$= \exp(-\frac{z_{-1}^2}{2})\left(-A \cdot \frac{\eta}{2}\right)(1 + \frac{1}{\sigma})\left(1 - \frac{Ag(\alpha)}{\sqrt{A^2 g^2(\alpha) + h(\sigma)}}\right) < 0, \quad (35)$$

where $z_{+1} = -Ag(\alpha) + \sqrt{\left(\frac{A}{\sigma}\right)^2 g^2(\alpha) + \frac{h(\sigma)}{\sigma^2}}$ and $z_{-1} = \frac{A}{\sigma}g(\alpha) - \sqrt{A^2 g^2(\alpha) + h(\sigma)}$. We leverage the property that $z_{+1}^2 < z_{-1}^2$ given $\sigma > 1$, and the property that the derivative of $z_{+1}$ with respect to $\alpha$ is less than zero, which is demonstrated in Theorem 1. Hence, as $\alpha$ increases, the discrepancy in robust errors between the two classes in the robust model diminishes monotonically. For the standard error of the standard model, since $0 < \epsilon < \eta$, substituting $g(\alpha) = \frac{1+\alpha}{2}\eta - \epsilon$ for $g(\alpha) = \frac{1+\alpha}{2}\eta$ in the proof of Theorem 2 doesn't alter the outcome of the error's derivative with respect to $\alpha$. Hence, as $\alpha$ increases, the discrepancy in standard errors between the two classes in the standard model diminishes monotonically.

$\square$

## C.2 ANALYSIS ON THE ROBUST FAIRNESS PROBLEM

In Sections 3.1 and 3.2, we demonstrate that robust fairness deteriorates as the distance between classes becomes closer. In this context, we examine the correlation between inter-class distance and the robust fairness issue, which is the phenomenon that accuracy disparity amplifies when transitioning from a standard to an adversarial setting. From the results of Fig 3c, we can observe that as the distance between classes increases, the difference between the results of standard and robust settings also decreases. This suggests that the robust fairness issue also can be exacerbated by similar classes. To investigate this theoretically, we will prove the following corollary, drawing on the proof from Theorem 2. Let's denote $f^{\alpha_1}$ as the optimal classifier when $\alpha$ in equation 2 is set to $\alpha = \alpha_1$. The following corollary establishes the relationship between the robust fairness problem and the distance between classes.

**Corollary 1.** *When given two different classification tasks with different alpha values of $\alpha_1$ and $\alpha_2$ (where $\alpha_1 < \alpha_2$) and $dg^2(\alpha) \gg \sigma$, the following inequality holds:*

$$D\big(\mathcal{R}_{rob}(f_{rob}^{\alpha_2})\big) - D\big(\mathcal{R}_{nat}(f_{nat}^{\alpha_2})\big) < D\big(\mathcal{R}_{rob}(f_{rob}^{\alpha_1})\big) - D\big(\mathcal{R}_{nat}(f_{nat}^{\alpha_1})\big). \tag{36}$$

*Proof.* We aim to prove that, given the difference in the value of $g(\alpha)$ by $\epsilon$ (representing the input distribution difference between a standard setting and an adversarial setting), the difference in the output of function $D$ increases as $\alpha$ decreases. In other words, the above inequality implies that as $\alpha$ decreases, the slope of the function $D$ becomes steeper. We demonstrated in Theorem 2 that the function $D$ is a decreasing function with respect to $\alpha$. Therefore, if function $D$ is proven to be a convex function, then the inequality holds true. We assume that the dimension $d$ of the input notably surpasses the variance of the class $y = +1$, and that, the distance between the centers of the two distributions remains sufficiently vast to ensure learning in an adversarial setting, satisfying $dg^2(\alpha) \gg \sigma$ both in standard and adversarial settings. Then, the second derivative of function $D$ with respect to $\alpha$, $\frac{\partial^2}{\partial \alpha^2}(D(g(\alpha)))$, is as follows:

$$\exp\big(-\frac{z_{+1}^2}{2}\big)(-z_{+1})(z_{+1}')^2 + \exp\big(-\frac{z_{+1}^2}{2}\big)(z_{+1}'') - \exp\big(-\frac{z_{-1}^2}{2}\big)(-z_{-1})(z_{-1}')^2 - \exp\big(-\frac{z_{-1}^2}{2}\big)(z_{-1}''), \tag{37}$$

where $z_{+1} = -Ag(\alpha) + \sqrt{\left(\frac{A}{\sigma}\right)^2 g^2(\alpha) + \frac{h(\sigma)}{\sigma^2}}$, $z_{-1} = \frac{A}{\sigma}g(\alpha) - \sqrt{A^2 g^2(\alpha) + h(\sigma)}$, $z_{+1}' = \frac{\partial z_{+1}}{\partial \alpha}$, and $z_{-1}' = \frac{\partial z_{-1}}{\partial \alpha}$, $z_{+1}'' = \frac{\partial^2 z_{+1}}{\partial \alpha^2}$, and $z_{-1}'' = \frac{\partial^2 z_{-1}}{\partial \alpha^2}$. We will now demonstrate that equation 37 has a value greater than zero. First, $z_{+1}''$ and $z_{-1}''$ are given as follows:

$$z_{+1}'' = \left(\frac{\left(\frac{A}{\sigma}\right)^2 g(\alpha)}{\sqrt{\left(\frac{A}{\sigma}\right)^2 g^2(\alpha) + \frac{h(\sigma)}{\sigma^2}}}\right)' \cdot \frac{\eta}{2} = \left(\frac{\left(\frac{A}{\sigma}\right)^2 \frac{h(\sigma)}{\sigma^2}}{\left(\left(\frac{A}{\sigma}\right)^2 g^2(\alpha) + \frac{h(\sigma)}{\sigma^2}\right)^{\frac{3}{2}}}\right) \cdot \frac{\eta^2}{4} > 0, \tag{38}$$

$$z_{-1}'' = \left(-\frac{A^2 g(\alpha)}{\sqrt{A^2 g^2(\alpha) + h(\sigma)}}\right)' \cdot \frac{\eta}{2} = -\sigma\left(\frac{\left(\frac{A}{\sigma}\right)^2 \frac{h(\sigma)}{\sigma^2}}{\left(\left(\frac{A}{\sigma}\right)^2 g^2(\alpha) + \frac{h(\sigma)}{\sigma^2}\right)^{\frac{3}{2}}}\right) \cdot \frac{\eta^2}{4} < 0. \tag{39}$$

Consequently, $\exp\big(-\frac{z_{+1}^2}{2}\big)(z_{+1}'') > 0$ and $-\exp\big(-\frac{z_{-1}^2}{2}\big)(z_{-1}'') > 0$. Next, in order to prove that $\exp\big(-\frac{z_{+1}^2}{2}\big)(-z_{+1})(z_{+1}')^2 - \exp\big(-\frac{z_{-1}^2}{2}\big)(-z_{-1})(z_{-1}')^2 > 0$, we first calculate $(z_{+1}z_{+1}') \cdot \frac{2}{\eta}$ considering that $dg^2(\alpha) \gg \sigma$.

$$(z_{+1}z_{+1}') \cdot \frac{2}{\eta} = Ag(\alpha) + \frac{A}{\sigma^2}g(\alpha) - \frac{1}{\sigma}\sqrt{A^2 g^2(\alpha) + h(\sigma)} - \frac{1}{\sigma}\frac{A^2 g^2(\alpha)}{\sqrt{A^2 g^2(\alpha) + h(\sigma)}}$$

$$= Ag(\alpha) + \frac{A}{\sigma^2}g(\alpha) - \frac{1}{\sigma}\left(\frac{2A^2 g^2(\alpha) + h(\sigma)}{\sqrt{A^2 g^2(\alpha) + h(\sigma)}}\right) > Ag(\alpha) + \frac{A}{\sigma^2}g(\alpha) - \frac{1}{\sigma}\left(\frac{2A^2 g^2(\alpha) + h(\sigma)}{\sqrt{A^2 g^2(\alpha)}}\right)$$

$$= Ag(\alpha)(1-\frac{1}{\sigma})^2 - \frac{1}{\sigma} \cdot \frac{h(\sigma)}{Ag(\alpha)} \xrightarrow{\times (\sigma+1)^2(\sigma-1)Ag(\alpha)} 4dg^2(\alpha)(\sigma-1) - 2\sigma(\sigma+1)\log\sigma > 0.$$

$$\tag{40}$$

Calculating $z_{-1}z'_{-1}$ yields the same value as $z_{+1}z'_{+1}$. Taking into account Theorem 2, where it's established $z_{-1} < 0$, $z'_{+1} < 0$, and $z^2_{-1} > z^2_{+1}$, the expression of the equation, $\exp\left(-\frac{z^2_{+1}}{2}\right)(-z_{+1})(z'_{+1})^2 - \exp\left(-\frac{z^2_{-1}}{2}\right)(-z_{-1})(z'_{-1})^2$, is given by:

$$\exp\left(-\frac{z^2_{+1}}{2}\right)(-z_{+1})(z'_{+1})^2 - \exp\left(-\frac{z^2_{-1}}{2}\right)(-z_{-1})(z'_{-1})^2 > \exp\left(-\frac{z^2_{+1}}{2}\right)(-z_{+1}z'^2_{+1} + z_{-1}z'^2_{-1})$$
$$= \frac{\eta}{2}\exp\left(-\frac{z^2_{+1}}{2}\right)(z_{+1}z'_{+1})\left(A - \frac{A}{\sigma} + \frac{A^2 g(\alpha)}{\sqrt{A^2 g^2(\alpha) + h(\sigma)}} - \frac{1}{\sigma}\cdot\frac{A^2 g(\alpha)}{\sqrt{A^2 g^2(\alpha) + h(\sigma)}}\right) > 0.$$
(41)

As a result, the following inequality is satisfied.

$$\frac{\partial^2}{\partial\alpha^2}(D(g(\alpha))) = \exp\left(-\frac{z^2_{+1}}{2}\right)(-z_{+1})(z'_{+1}) + \exp\left(-\frac{z^2_{+1}}{2}\right)(z''_{+1})$$
$$- \exp\left(-\frac{z^2_{-1}}{2}\right)(-z_{-1})(z'_{-1}) - \exp\left(-\frac{z^2_{-1}}{2}\right)(z''_{-1}) > 0. \quad (42)$$

This indicates that the function $D$ is a convex function with respect to $\alpha$. Consequently, equation 36 is proven.

$\square$

### C.3 THEORETICAL ANALYSIS ON ADVERSARIAL MARGIN

In this section, we conduct theoretical analyses on the characteristics that emerge when different values of the adversarial margin, $\epsilon$, are allocated to distinct classes during adversarial training.

**A binary classification task for adversarial margin** $\epsilon$ The setup of the binary classification task is similar to that of Section 3.1. However, the two classes in this task have identical variances and the value of $\alpha$ is set to $\alpha = 1$.

$$y \overset{u.a.r}{\sim} \{-1, +1\}, \ \boldsymbol{\mu} = (\eta, \eta, \cdots, \eta), \qquad \boldsymbol{x} \sim \begin{cases} \mathcal{N}(\boldsymbol{\mu}, 1), & \text{if } y = +1 \\ \mathcal{N}(-\boldsymbol{\mu}, 1), & \text{if } y = -1 \end{cases} \quad (43)$$

Here, we assume that adversarial attacks, created with different margins for each class under the adversarial setting, are applied. Specifically, the upper bound of the attack margin for class $y = -1$ (denoted as $\epsilon_{-1}$) is assumed to be smaller than the upper bound of the attack margin for class $y = +1$ (denoted as $\epsilon_{+1}$). In these conditions, the following theorem holds:

**Theorem 3.** *Let $0 < \epsilon_{-1} < \epsilon_{+1} < \eta$. Then, the adversarial training of the classifier $f$ will increase the standard error for class $y = +1$ while decreasing the standard error for class $y = -1$.*

$$\begin{aligned} \mathcal{R}_{nat}(f_{nat}| - 1) &< \mathcal{R}_{nat}(f_{rob}| - 1), \\ \mathcal{R}_{nat}(f_{nat}| + 1) &> \mathcal{R}_{nat}(f_{rob}| + 1). \end{aligned} \quad (44)$$

*Proof.* Based on Lemma 1, when $\|\boldsymbol{w}\|_2 = 1$, the optimal parameters for the standard classifier are given by $\boldsymbol{w} = (\frac{1}{\sqrt{d}}, \frac{1}{\sqrt{d}}, ..., \frac{1}{\sqrt{d}})$. Because the distributions in the task have identical variances, the optimal parameter $b$ is set to $b = 0$. As a result, the standard error of the standard classifier for the class $y = -1$ is equivalent to that for the class $y = +1$. In adversarial settings, however, adversarial attacks cause the centroid of each class to move closer to the centroid of its counterpart. Per Lemma 2, for datasets affected by such class shifts, the decision boundary is defined by the mean of the centroids of the two classes. Given that $\frac{\epsilon_{-1} - \epsilon_{+1}}{2} < 0$, the decision boundary lies nearer to the centroid of the class $y = -1$. Consequently, $\mathcal{R}_{nat}(f_{rob}| + 1)$ yields a value less than $\mathcal{R}_{nat}(f_{nat}| + 1)$, while class $y = -1$ exhibits the opposite outcome. $\square$

Theorem 3 implies that in adversarial training, when each class is trained with a distinct adversarial margin, the class subjected to an attack with a relatively higher upper bound for the adversarial margin will be trained to reduce its standard error.

Let's denote the robust error under an adversarial attack with an upper bound of $\epsilon$ for the test set as $\mathcal{R}_{rob,\epsilon}$, and let's denote $f_{rob,\epsilon}$ as the classifier trained with an adversarial margin of $\epsilon$ applied equally to both classes $y = +1$ and $y = -1$. In this context, the robust classifier trained with different attack margins $\epsilon_{-1}$ and $\epsilon_{+1}$ adheres to the following corollary:

**Corollary 2.** *Let the upper bounds of adversarial margins for the above classification task satisfy* $0 < \epsilon_{-1} < \epsilon_{+1} < \eta$, *and let the upper bound of adversarial margin for the test adversarial attack satisfy* $0 < \epsilon < \eta$, *and be applied uniformly across both classes. Then, it satisfies*

$$\mathcal{R}_{rob,\epsilon}(f_{rob}| - 1) - \mathcal{R}_{rob,\epsilon}(f_{rob,\epsilon}| - 1) > \mathcal{R}_{rob,\epsilon}(f_{rob}| + 1) - \mathcal{R}_{rob,\epsilon}(f_{rob,\epsilon}| + 1). \tag{45}$$

Because the optimal parameter of $f_{rob,\epsilon}$ is the same as that for $f_{nat}$, Corollary 2 can be proven using the same method as Theorem 3. This corollary indicates that during adversarial training, classes trained with a larger value of $\epsilon$ can exhibit a smaller increase in error compared to those trained with a lesser value of $\epsilon$. This conclusion is supported both by the aforementioned theoretical analysis and the empirical evidence shown in Fig. 5. Assigning a higher adversarial margin to the harder class, in comparison to other classes, helps improve robust fairness.

# D    ADDITIONAL IMPLEMENTATION AND EXPERIMENTAL DETAILS

## D.1    IMPLEMENTATION DETAILS

**Datasets**    The CIFAR-10 dataset (Krizhevsky et al., 2009) contains 50,000 training images and 10,000 test images, spanning 10 classes. Similarly, CIFAR-100 (Krizhevsky et al., 2009) comprises 50,000 training images and 10,000 test images distributed across 100 classes. Both CIFAR-10 and CIFAR-100 have image dimensions of $32 \times 32$ pixels. These datasets are subsets of the 80 million tiny images dataset (Torralba et al., 2008), which aggregates images retrieved from seven distinct image search engines. The STL-10 dataset (Coates et al., 2011) provides 5,000 training images and 8,000 test images across 10 classes, with each image having dimensions of $96 \times 96$ pixels. For our experiments, we resized the STL-10 dataset images to $64 \times 64$ pixels during model training. More details on these datasets are available at `https://paperswithcode.com/datasets`.

**Implementation details on experiments**    We performed experiments on the CIFAR-10, CIFAR-100 (Krizhevsky et al., 2009), and STL-10 (Coates et al., 2011) datasets. As the baseline adversarial training algorithm, we employed TRADES (Zhang et al., 2019). Our models utilized the ResNet18 architecture (He et al., 2016b). We set the learning rates to 0.1, implementing a decay at the 100th and 105th epochs out of a total of 110 epochs, using a decay factor of 0.1 as recommended by Pang et al. (2021). For optimization, we utilized stochastic gradient descent with a weight decay factor of 5e-4 and momentum set to 0.9. The upper bounds for adversarial perturbation were determined at 0.031 ($\epsilon = 8$). The step size for generating adversarial examples for each model was set to one-fourth of the $\ell_\infty$-bound of the respective model, over a span of 10 steps. To assess the robustness of the models, we employed the adaptive auto attack ($A^3$) method (Liu et al., 2022) for a rigorous evaluation. Furthermore, we adopted the evaluation metric $\rho$ as proposed by Li & Liu (2023), which is defined as follows:

$$\rho(f, \Delta, \mathcal{A}) = \frac{\text{Acc}_{wc}(\mathcal{A}_\Delta(f)) - \text{Acc}_{wc}(\mathcal{A}(f))}{\text{Acc}_{wc}(\mathcal{A}(f))} - \frac{\text{Acc}(\mathcal{A}_\Delta(f)) - \text{Acc}(\mathcal{A}(f))}{\text{Acc}(\mathcal{A}(f))} \tag{46}$$

$\text{Acc}(\cdot)$ represents average accuracy, while $\text{Acc}_{wc}(\cdot)$ denotes the worst-class accuracy. $\mathcal{A}(f)$ symbolizes the model $f$ trained using the baseline algorithm, and $\mathcal{A}_\Delta(f)$ indicates the model $f$ trained employing a different algorithm that utilizes strategy $\Delta$. This metric allows us to evaluate whether the employed algorithm enhances the worst accuracy performance while maintaining average accuracy performance relative to the baseline algorithm. A higher value of $\rho$ suggests superior method performance. Experiments were executed three times using three distinct seeds, with the results reflecting the average value across the final five epochs. To assess the performance of other methods, we made use of the original code for each respective methodology. For each methodology's specific

hyperparameters, we utilized the hyperparameters from the original code for the CIFAR-10 dataset. For other datasets, we conducted experiments with various hyperparameter settings, including those from the original code. FRL (Xu et al., 2021) involves 40 epochs of fine-tuning after pretraining, WAT utilizes the 75-90-100 LR scheduling identical to the settings of TRADES (Zhang et al., 2019), and both CFA (Wei et al., 2023) and BAT adopt the 100-150-200 LR scheduling following the settings of PGD (Madry et al., 2017). As the performance difference between ResNet18 (He et al., 2016b) and PreActResNet18 (He et al., 2016a) is not significant, we proceeded with the comparison as is. We measured the average performance of the five checkpoints after the second LR decay for CFA, BAT, and WAT. For FRL, using the pretrained model provided by the official code GitHub (`https://github.com/hannxu123/fair_robust`), we conducted model training with 40 epochs of fine-tuning using PreActResNet18 architecture. We reported the results that showed the highest worst robust accuracy. Our experiments were carried out on a single RTX 8000 GPU equipped with CUDA11.6 and CuDNN7.6.5.

## D.2 EXPERIMENTAL DETAILS

**Experimental details of Fig. 1** In Fig. 1, the central portion illustrates the class-wise accuracy of the TRADES model (Zhang et al., 2019), accompanied by the prediction outcomes of the cat class, using test data from the CIFAR-10 dataset. We assessed the performance across classes utilizing $A^3$ (Liu et al., 2022). The results depict the accuracy variations of each model in comparison to the TRADES model. Performance results for the previous method were derived from the FRL method (Xu et al., 2021).

Specifically, the red and blue arrows next to "Hard," "Easy," "Similar," and "Dissimilar" in Fig. 1 represent the facilitation or restriction of learning for the respective classes or class sets. When pointing upwards, it indicates the facilitation of learning for the associated class, whereas pointing downwards signifies a restriction of the learning for that class. The arrow size corresponds to the degree of facilitation or restriction. The color of the arrows is simply for differentiation, and the roles of the arrows are the same. The arrow for the class "cat" points upwards, denoting the facilitation of learning. This is because the cat is considered the worst class, and to improve robust fairness, learning for the cat class is promoted by restricting the learning of other classes. The conventional approach ("Hard" and "Easy") restricts hard classes less (small arrows) and easy classes more (large arrows) with reference to class-wise accuracy (red bar). In contrast, our proposed method, considering the relative difficulty between classes and incorporating class-wise similarity with reference to the prediction of the cat class (blue bar), introduces new blue arrows alongside the existing red arrows. Similar classes to the cat receive additional downward blue arrows beyond the small red arrows, as they are more restricted compared to the conventional difficulty-based approach. Conversely, dissimilar classes to the cat receive additional upward blue arrows beyond the red arrows, as they are less restricted compared to the conventional approach.

**Experimental details of Fig. 2** In Fig, 2, we trained two distinct models using the TRADES method. One model was trained on five classes from CIFAR-10 (cat, horse, deer, dog, bird), while the other was trained on another set of five classes (cat, plane, ship, truck, car). For evaluation, we assessed the performance of each class using $A^3$ (Liu et al., 2022). All other implementation details were kept consistent with the previously described experiments.

**Experimental details of Fig. 3** In the results presented in Fig. 3a, we quantified the distance between classes using models trained via TRADES (Zhang et al., 2019). This distance was determined using the feature output of the TRADES model, specifically leveraging the penultimate layer's output as the feature representation. We designated the median feature of each class as its distribution's center and computed distances between these central features. In our analyses for Figs. 3b and 3c, we identified the bird class as the hard class due to its consistently lower accuracy in most CIFAR-10 binary classification tasks, especially when paired with the three other classes: deer, horse, and truck. These three classes exhibit nearly identical accuracies when paired with each other in a binary classification, suggesting that their difficulties or variances are also almost equivalent. All models were trained using the TRADES methodology. Standard classifiers were assessed using clean samples, while robust classifiers underwent evaluation via the PGD attack (Madry et al., 2017), which employed an adversarial margin of $\epsilon = 8$, a step number of 20, and a step size of 0.765.

**Experimental details of Fig. 4**  We measured class-wise accuracy, variance, and distance using models trained with TRADES (Zhang et al., 2019). The class-wise accuracy was determined by evaluating the performance of each class using $A^3$ (Liu et al., 2022). Both the class-wise variance and distance were assessed through the feature output of the TRADES model, where we utilized the output of the penultimate layer as the feature output. Given a trained model, the variance and distance of the distribution for each class can be defined as follows: Let $g(x)$ represent the feature output of the trained model, $N_c$ be the number of data in class $c$, and $x_{i,c}$ be the $i$-th data of class $c$. Then, the central feature of class $c$, denoted as $\boldsymbol{\mu}_c$, is given by $\boldsymbol{\mu}_c = \frac{1}{N_c}\sum_i^{N_c} g(x_{i,c})$. Subsequently, the variance is defined as $\sigma_c^2 = \frac{1}{N_c}\sum_i^{N_c}(g(x_{i,c}) - \boldsymbol{\mu}_c)^2$. For two different classes, $c_1$ and $c_2$, the distance is defined as $Distance(c_1, c_2) = \|\boldsymbol{\mu}_{c_1} - \boldsymbol{\mu}_{c_2}\|_2$. Using these definitions, we can approximately measure the variance within each class and the distance between classes that the model has learned.

**Experimental details of Fig. 5**  Fig. 5 displays the results of three models trained on a binary classification task to distinguish between the bird and cat classes from CIFAR-10 using the TRADES methodology. Each model was trained with the adversarial margins depicted in the figure, and robust accuracy evaluation was performed using the projected gradient ascent attack (Madry et al., 2017). The evaluation attack has an adversarial margin of $\epsilon = 6$, a step number of 20, and a step size of 0.57375.

## E  ABLATION STUDIES

Table 3: Performance of DAFA across various architectures on the CIFAR-10 dataset. The best results are indicated in bold.

| Arch. | Method | Clean | | Robust | |
|---|---|---|---|---|---|
| | | Average | Worst | Average | Worst |
| PRN18 | TRADES | $81.93 \pm 0.17$ | $66.41 \pm 1.12$ | $\mathbf{49.55 \pm 0.15}$ | $21.50 \pm 1.51$ |
| | + DAFA$_{\mathcal{L}_{CE}}$ | $\mathbf{81.98 \pm 0.23}$ | $\mathbf{69.03 \pm 1.27}$ | $49.10 \pm 0.18$ | $28.49 \pm 1.25$ |
| | + DAFA$_\epsilon$ | $81.93 \pm 0.31$ | $66.71 \pm 1.26$ | $49.38 \pm 0.19$ | $23.65 \pm 1.19$ |
| | + DAFA | $81.56 \pm 0.30$ | $67.66 \pm 1.19$ | $48.72 \pm 0.25$ | $\mathbf{30.89 \pm 1.32}$ |
| WRN28 | TRADES | $\mathbf{85.70 \pm 0.13}$ | $71.97 \pm 0.90$ | $\mathbf{53.95 \pm 0.12}$ | $26.24 \pm 0.85$ |
| | + DAFA$_{\mathcal{L}_{CE}}$ | $85.62 \pm 0.19$ | $\mathbf{73.11 \pm 0.89}$ | $53.30 \pm 0.26$ | $30.31 \pm 1.30$ |
| | + DAFA$_\epsilon$ | $85.14 \pm 0.18$ | $71.79 \pm 1.10$ | $53.49 \pm 0.25$ | $28.81 \pm 1.27$ |
| | + DAFA | $84.94 \pm 0.22$ | $70.91 \pm 0.98$ | $52.99 \pm 0.21$ | $\mathbf{34.24 \pm 1.03}$ |

**Ablation study on other architectures**  Table 3 illustrates the effectiveness of our methodology when applied to different model architectures: PreActResNet18 (PRN18) (He et al., 2016b) and WideResNet28 (WRN28) (Zagoruyko & Komodakis, 2016). The results indicate that the application of our methodology consistently enhances the worst robust accuracy compared to the baseline models, thereby reducing accuracy disparity. Both class-wise components in the TRADES model, the cross-entropy loss $\mathcal{L}_{CE}$ and adversarial margin $\epsilon$, addressed through DAFA, demonstrate efficacy in improving robust fairness. Their combined application showcases the most significant effect.

Table 4: Performance of DAFA for varying warm-up epoch on the CIFAR-10 dataset.

| Warm-up | Clean | | Robust | |
|---|---|---|---|---|
| | Average | Worst | Average | Worst |
| 10 | $\mathbf{81.94 \pm 0.24}$ | $\mathbf{68.83 \pm 1.36}$ | $48.92 \pm 0.26$ | $29.59 \pm 1.30$ |
| 30 | $81.73 \pm 0.18$ | $67.97 \pm 1.07$ | $48.77 \pm 0.29$ | $29.32 \pm 1.23$ |
| 50 | $81.52 \pm 0.37$ | $67.67 \pm 1.23$ | $48.89 \pm 0.21$ | $\mathbf{30.58 \pm 0.74}$ |
| 70 | $81.63 \pm 0.14$ | $67.90 \pm 1.25$ | $49.05 \pm 0.14$ | $30.53 \pm 1.04$ |
| 90 | $81.58 \pm 0.26$ | $67.95 \pm 0.94$ | $\mathbf{49.06 \pm 0.13}$ | $30.48 \pm 0.97$ |

**Ablation study on hyperparameter of DAFA**    Table 4 displays the performance when the warm-up epoch of DAFA is altered. The table suggests that, for warm-up epochs of 50 or more, the performance remains largely consistent. When calculating the class-wise weight $\mathcal{W}$ for applying DAFA at lower epochs, the improvement in worst robust accuracy is relatively modest. However, the results still reflect a significant boost in worst robust accuracy with minimal average accuracy degradation compared to existing methods.

Table 5: Performance of DAFA for varying $\lambda$ on the CIFAR-10 dataset.

| $\lambda$ | Clean | | Robust | |
| --- | --- | --- | --- | --- |
| | Average | Worst | Average | Worst |
| 0.5 | **82.13 ± 0.20** | **69.07 ± 1.00** | **49.52 ± 0.26** | 26.07 ± 1.26 |
| 1.0 | 81.63 ± 0.14 | 67.90 ± 1.25 | 49.05 ± 0.14 | 30.53 ± 1.04 |
| 1.5 | 80.87 ± 0.21 | 64.51 ± 1.39 | 48.36 ± 0.14 | **32.00 ± 0.92** |
| 2.0 | 79.80 ± 0.29 | 62.16 ± 1.87 | 47.39 ± 0.23 | 29.27 ± 1.64 |

Table 5 presents the performance when altering the hyperparameter $\lambda$ in DAFA. As the value of $\lambda$ increases, the variation in class-wise weight $\mathcal{W}$ also enlarges, leading to significant performance shifts. At all values of $\lambda$, an enhancement in worst robust accuracy is observed, with the results at $\lambda = 1.5$ showing a stronger mitigation effect on the fairness issue than the results at $\lambda = 1.0$.

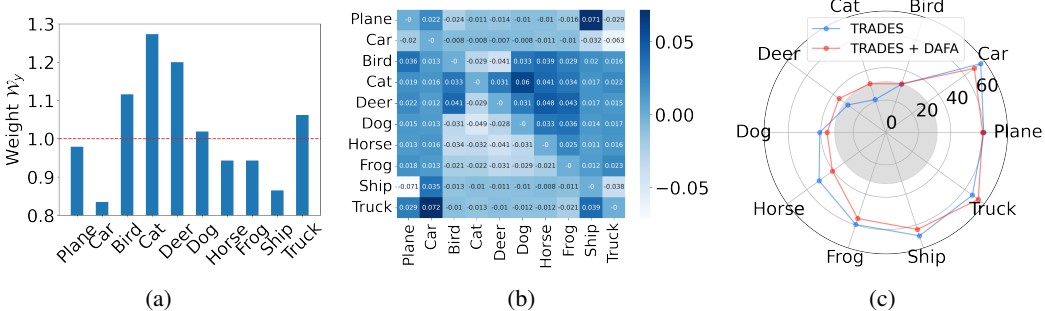

| (a) | (b) | (c) |
| --- | --- | --- |

Figure 7: (a) illustrates the weight $\mathcal{W}$ for each class on the CIFAR-10 dataset, computed using the DAFA methodology. (b) presents the matrix showing the variations in weight $\mathcal{W}$ that was reduced for each class through the application of DAFA. (c) presents a comparison of the class-wise accuracy for the CIFAR-10 dataset using the TRADES model, both before and after the application of DAFA.

**Detailed results of DAFA**    Fig. 7a depicts the class-wise weights $\mathcal{W}$ determined after applying DAFA to the TRADES methodology on the CIFAR-10 dataset. Observing the figure, it's evident that hard classes tend to exhibit higher values compared to their similar classes. Fig. 7b illustrates a matrix that reveals the values of weight shifts $\mathcal{W}$ between each class pair when applying DAFA to the TRADES methodology. In essence, the weight $\mathcal{W}$ shifts between similar classes are notably larger compared to the shifts between dissimilar class pairs. Fig. 7c presents the class-wise robust accuracy for the TRADES model and a model trained by applying DAFA to the former. From the figure, it's clear that our approach enhances the performance of hard classes, thereby mitigating fairness issues in adversarial trianing.

**Application of DAFA to other adversarial training algorithms**    In addition to the TRADES method, we conducted experiments on applying DAFA to various adversarial training algorithms. Table 6 presents the results of applying DAFA to standard adversarial training using Projected Gradient Descent (PGD) (Madry et al., 2017). The results demonstrate that applying DAFA to the PGD model effectively enhances worst robust accuracy while maintaining average robust accuracy. However, when DAFA is applied solely to the cross-entropy loss $\mathcal{L}_{CE}$, it outperforms results that employ both $\mathcal{L}_{CE}$ and the adversarial margin $\epsilon$. Furthermore, applying DAFA only to $\epsilon$ results in a lower worst robust accuracy than the baseline.

In Appendix C.3, we already demonstrated that, within an adversarial setting, when training with different values of $\epsilon$, a class trained with a larger $\epsilon$ exhibits higher worst robust accuracy. However, since we observed the opposite outcome for the PGD model, we empirically re-examine the effect of $\epsilon$ on the PGD model through a simple experiment. Fig. 8 presents the results of the empirical verification carried out on the PGD model, akin to what was performed for the TRADES model in Fig. 5. As in the experiment of Fig. 5, each model was trained with distinct adversarial margins for each class, yet all were tested using a consistent adversarial margin of $\epsilon = 6$. The results clearly indicate that, similar to the findings in Fig. 5, both clean and robust accuracy metrics showcase superior performance for the class trained with the relatively larger $\epsilon$. This outcome is consistent

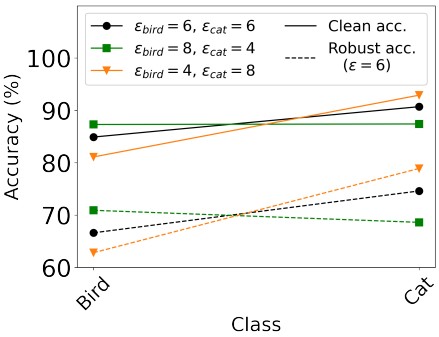

Figure 8: The results from an experiment on the PGD (Madry et al., 2017) model, mirroring the conditions of Fig. 5.

with our assertion that to enhance the performance of a hard class, it should be allocated a larger value of $\epsilon$. Consequently, while our theoretical insights apply to both the TRADES model and the binary classification PGD model, they do not apply to the multi-class classification PGD model.

Table 6: Performance results of applying DAFA to PGD on the CIFAR-10 dataset.

| Method | Clean | | Robust | |
|---|---|---|---|---|
| | Average | Worst | Average | Worst |
| PGD | $84.66 \pm 0.13$ | $64.55 \pm 1.41$ | $48.65 \pm 0.14$ | $18.27 \pm 1.35$ |
| PGD + DAFA$_{\mathcal{L}_{CE}}$ | $\mathbf{84.68 \pm 0.23}$ | $\mathbf{69.11 \pm 1.39}$ | $48.53 \pm 0.26$ | $\mathbf{25.45 \pm 1.31}$ |
| PGD + DAFA$_\epsilon$ | $83.12 \pm 0.18$ | $54.91 \pm 1.53$ | $\mathbf{48.94 \pm 0.09}$ | $15.24 \pm 0.90$ |
| PGD + DAFA | $83.93 \pm 0.21$ | $67.40 \pm 1.35$ | $48.02 \pm 0.28$ | $21.83 \pm 1.39$ |

A plausible explanation for the observed phenomenon is the presence or absence of learning from the clean data distribution. While the PGD model learns solely from adversarial examples, the TRADES model utilizes both clean and adversarial examples. Intuitively, the reason for the disparity between the results of the binary classification PGD model and the multi-class classification PGD model might be attributed to the number of opposing classes. In the binary classification PGD model, there's only one opposing class; hence, even if it learns only from adversarial examples, it can still maintain the vicinity around the center of the clean data distribution. However, in the multi-class classification PGD model, with multiple opposing classes, solely learning from adversarial examples might result in the vicinity around the center of the clean data distribution being encroached upon by a third class. Indeed, looking at the clean accuracy of DAFA$_\epsilon$ for the worst class in Table 6, we can confirm a significant performance drop compared to the baseline model.

To validate the observations discussed above, we conducted experiments applying a methodology wherein the PGD model also learns around the center of the clean data distribution. Adversarial Vertex Mixup (AVmixup) (Lee et al., 2020) is an adversarial training approach that, instead of using adversarial examples, employs a randomly sampled data point from the linear interpolation of a clean data $x$ and its corresponding adversarial example $x + \delta$, denoted as $x + \alpha\delta$, where $0 \leq \alpha \leq 1$. The label $y$ uses the one-hot label without applying mixup, ensuring that the distribution from the center of the clean data to the adversarial example is learned. As shown in Table 6, when applying DAFA to the PGD model trained through AVmixup, the worst robust accuracy improves by 2%p compared to the baseline model. In conclusion, for models like TRADES that incorporate clean data into their training, assigning a larger value of $\epsilon$ to the hard class is a viable method to enhance the worst accuracy. However, in cases without such training, applying a large value of $\epsilon$ to the hard classes may not yield the desired results.

Additionally, we explored the application of our approach to another adversarial training algorithm, adversarial weight perturbation (AWP) (Wu et al., 2020). The setting is the same as that of TRADES for ResNet18. The results in Table 8 demonstrate that applying our method to AWP significantly improves robust worst accuracy.

Table 7: Performance results of applying DAFA$_\epsilon$ to PGD with the AVmixup (Lee et al., 2020) approach on the CIFAR-10 dataset.

| Method | Clean | | Robust | |
|---|---|---|---|---|
| | Average | Worst | Average | Worst |
| PGD + AVmixup | $\mathbf{89.56 \pm 0.18}$ | $\mathbf{77.63 \pm 1.54}$ | $41.64 \pm 0.39$ | $14.47 \pm 1.13$ |
| PGD + AVmixup + DAFA$_\epsilon$ | $89.03 \pm 0.17$ | $74.59 \pm 1.44$ | $\mathbf{41.82 \pm 0.18}$ | $\mathbf{16.37 \pm 1.23}$ |

Table 8: Performance results of applying DAFA to AWP (Wu et al., 2020) on the CIFAR-10 dataset.

| Method | Clean | | Robust | |
|---|---|---|---|---|
| | Average | Worst | Average | Worst |
| AWP | $\mathbf{81.33 \pm 0.02}$ | $64.30 \pm 1.10$ | $\mathbf{49.97 \pm 0.18}$ | $20.65 \pm 0.45$ |
| AWP + DAFA | $80.12 \pm 0.07$ | $\mathbf{66.74 \pm 1.35}$ | $49.42 \pm 0.16$ | $\mathbf{28.26 \pm 1.25}$ |

**Utilizing other types of class-wise similarity for DAFA**    Our method focuses on enhancing robust fairness by calculating class weights using class-wise similarity. Thus, our framework is not limited to prediction probability but can accommodate various types of class-wise similarity. Therefore, leveraging other class-wise similarities such as embedding similarity in our method is also feasible. To verify this, we employed the penultimate layer output as embedding vectors (e.g. 512-size vector for ResNet18). The cosine similarity between vectors for each class is computed, normalized, and subsequently applied to the DAFA framework. Let $e_i$ represent the mean embedding vector of class $i$. The similarity between class $i$ and class $j$ can be expressed as the normalized value of the cosine similarity between the two classes. In other words, if we denote $p_i(j)$ as the similarity between class $i$ and class $j$, then $p_i(j) = e_i \cdot e_j / Z$, where $Z = \sum_{k \in \{1, \dots, C\}} e_i \cdot e_k$. We applied the computed $p_i(j)$ to Eq. 7 in Section 4.2 and proceeded with the experiments. We chose $\lambda = 0.1$ since the variance of similarity values calculated using embeddings is not significant. Table 9 demonstrates that utilizing embedding similarity yields outcomes similar to the conventional results obtained using prediction probability. It is observed that embedding similarity also demonstrates effectiveness in improving robust fairness, as indicated by the results.

Theoretically, the value of $\mathcal{W}$ can be negative when using hard examples as a reference. This occurs in the most extreme case, where in Eq. 6, $i$ represents the easiest class, and all other classes $j$ are almost certain to be classified as class $i$, resulting in $\min \mathcal{W}_i \approx \mathcal{W}_{i,0} - 9$ for a 10-class classification. The maximum value of $\mathcal{W}_i$ is achieved when class $i$ is the most challenging class, with $p_i(i) = 0$ and $\sum_j p_i(j) = 1$, yielding $\max \mathcal{W}_i = \mathcal{W}_{i,0} + 1$. To address this, we scale $\mathcal{W}$ for each dataset using the $\lambda$ value in Eq. 7, ensuring that $\mathcal{W}$ is adjusted. Furthermore, we apply clipping to ensure $\mathcal{W}$ remains greater than 0 ($\mathcal{W} = \text{maximum}(\mathcal{W}, K)$ where $K > 0$). In our experiments, we selected $\lambda$ values of 1.0 for the CIFAR-10 dataset and 1.5 for the CIFAR-100 and STL-10 datasets, resulting in $\mathcal{W}$ values around 1.0.

Additionally, we conducted experiments to verify the effectiveness of our method when referencing easy classes instead of hard classes. For our approach in Section 4.2, we restrict the learning of easy classes and promote the learning of hard classes using hard classes as a reference. However, irrespective of which class between $i$ and $j$ is relatively easy or hard, if $i$ and $j$ form a similar class pair, the probability of class $i$ being classified as $j$ and the probability of class $j$ being classified as $i$ both yield high values. Conversely, if they are dissimilar, both probabilities are low. Therefore, instead of using $p_i(j)$ with hard classes as a reference, we can use $p_j(i)$ with easy classes as a reference. Then, Eq. 7 in Section 4.2 is modified as $\mathcal{W}_i = \mathcal{W}_{i,0} + \sum_{j \neq i} 1[p_i(i) < p_j(j)] \cdot p_j(i) - 1[p_i(i) > p_j(j)] \cdot p_i(j)$. The experiment results when referencing easy classes instead of hard classes are illustrated in Table 9. Experimental results indicate that clean, average robust, and worst robust accuracy are all similar to the original results.

**Ablation study on hyperparameter of TRADES with DAFA**    We conducted experiments by varying the value of $\beta$ of the TRADES method and applying our method. We compared the enhanced robust fairness with the baseline TRADES model. Table 10 shows the results when the value of $\beta$

Table 9: Performance results of utilizing other class-wise similarities for DAFA on the CIFAR-10 dataset.

| Method | Clean | | Robust | |
|---|---|---|---|---|
| | Average | Worst | Average | Worst |
| TRADES | $82.04 \pm 0.10$ | $65.70 \pm 1.32$ | $49.80 \pm 0.18$ | $21.31 \pm 1.19$ |
| TRADES + DAFA | $81.63 \pm 0.08$ | $67.90 \pm 1.25$ | $49.05 \pm 0.14$ | $30.53 \pm 1.04$ |
| TRADES + DAFA (embedding) | $81.87 \pm 0.12$ | $69.80 \pm 1.35$ | $48.92 \pm 0.08$ | $29.60 \pm 1.27$ |
| TRADES + DAFA (easy ref.) | $81.71 \pm 0.08$ | $67.28 \pm 1.48$ | $49.08 \pm 0.21$ | $30.22 \pm 0.82$ |

varies (1.0, 3.0, 6.0, and 9.0). Although as the overall robust performance of the baseline models decreases, the robust performance of the models with DAFA also decreases, we can observe that, in all settings, the clean/average robust accuracy performance is maintained close to the baseline, while significantly improving the worst robust accuracy.

Table 10: Performance of DAFA for varying the value of $\beta$ for TRADES on the CIFAR-10 dataset.

| Method | Clean | | Robust | |
|---|---|---|---|---|
| | Average | Worst | Average | Worst |
| TRADES ($\beta = 1.0$) | $87.78 \pm 0.22$ | $74.70 \pm 1.49$ | $43.70 \pm 0.29$ | $16.00 \pm 1.15$ |
| TRADES ($\beta = 1.0$) + DAFA | $87.43 \pm 0.23$ | $75.22 \pm 1.15$ | $43.12 \pm 0.25$ | $20.62 \pm 1.14$ |
| TRADES ($\beta = 3.0$) | $85.03 \pm 0.19$ | $70.60 \pm 1.57$ | $48.64 \pm 0.35$ | $21.10 \pm 1.46$ |
| TRADES ($\beta = 3.0$) + DAFA | $84.58 \pm 0.14$ | $71.34 \pm 1.37$ | $47.68 \pm 0.16$ | $27.44 \pm 1.80$ |
| TRADES ($\beta = 6.0$) | $82.04 \pm 0.10$ | $65.70 \pm 1.32$ | $49.80 \pm 0.18$ | $21.31 \pm 1.19$ |
| TRADES ($\beta = 6.0$) + DAFA | $81.63 \pm 0.08$ | $67.90 \pm 1.25$ | $49.05 \pm 0.14$ | $30.53 \pm 1.04$ |
| TRADES ($\beta = 9.0$) | $79.94 \pm 0.13$ | $63.00 \pm 1.65$ | $49.85 \pm 0.24$ | $20.86 \pm 1.38$ |
| TRADES ($\beta = 9.0$) + DAFA | $79.49 \pm 0.10$ | $65.36 \pm 2.07$ | $48.86 \pm 0.17$ | $29.80 \pm 1.54$ |

## F  ADDITIONAL COMPARISON STUDIES

**Comparison study with other methods in each method's setting**  As explained in Appendix D, we conducted ablation studies comparing our methodology in each setting, given the different experimental setups for each methodology we compared. For the FRL setting, due to the low performance of the pretrained model, after 10 epochs of fine-tuning, we compute class weight and apply our method for the remaining 30 epochs. We measure the average performance of the last five checkpoints. Upon examining the results in Tables 11, 12, and 13, we observe that our method demonstrates average robust accuracy and clean accuracy comparable to the baseline algorithm in all settings. Simultaneously, it exhibits a high level of worst robust accuracy compared to existing methods.

**Comparison study with other methods in the same (our) setting**  Instead of comparing the performance of each method conducted in each setting, we conducted ablation studies comparing our methodology with other existing methods in the same setting. To assess the performance of other methods, we made use of the original code for each respective methodology. For each methodology's specific hyperparameters, we utilized the hyperparameters from the original code for the CIFAR-10 dataset. For other datasets, we conducted experiments with various hyperparameter settings, including those from the original code. We reported the results that showed the highest worst robust accuracy. However, we standardized default hyperparameters following the setting of Pang et al. (2021) as explained in D.1, such as adversarial margin, learning rate scheduling, model structure, and other relevant parameters. From Table 14, our method demonstrates average robust accuracy and clean accuracy comparable to the baseline algorithm in all datasets. Moreover, our method exhibits superior worst robust accuracy compared to existing methods.

Table 11: Comparison of the performance of the models for improving robust fairness methods in the setting of each method on the CIFAR-10 dataset: CFA (Wei et al., 2023), FRL (Xu et al., 2021), BAT (Sun et al., 2023), and WAT (Li & Liu, 2023). The best results among robust fairness methods are indicated in bold.

(a) CFA

| Method | Clean | | | Robust | | |
|---|---|---|---|---|---|---|
| | Average | Worst | $\rho_{nat}$ | Average | Worst | $\rho_{rob}$ |
| TRADES | $84.08 \pm 0.14$ | $67.22 \pm 2.22$ | - | $48.91 \pm 0.18$ | $20.42 \pm 1.69$ | - |
| + EMA | $83.97 \pm 0.15$ | $66.90 \pm 0.17$ | -0.01 | $50.62 \pm 0.10$ | $22.16 \pm 0.41$ | 0.12 |
| + CCM + CCR | $81.72 \pm 0.26$ | $65.84 \pm 2.07$ | -0.05 | $49.82 \pm 0.22$ | $22.58 \pm 1.72$ | 0.12 |
| + CCM + CCR + EMA | $81.68 \pm 0.17$ | $66.20 \pm 0.22$ | -0.08 | $\mathbf{51.01 \pm 0.05}$ | $23.16 \pm 0.10$ | 0.18 |
| + CCM + CCR + FAWA | $79.81 \pm 0.15$ | $65.18 \pm 0.16$ | -0.08 | $50.78 \pm 0.05$ | $27.62 \pm 0.04$ | 0.39 |
| + DAFA | $82.85 \pm 0.05$ | $68.48 \pm 0.68$ | 0.00 | $48.36 \pm 0.18$ | $29.68 \pm 0.77$ | 0.44 |
| + DAFA + EMA | $\mathbf{83.19 \pm 0.02}$ | $\mathbf{70.40 \pm 0.07}$ | $\mathbf{0.04}$ | $50.63 \pm 0.03$ | $29.95 \pm 0.11$ | 0.50 |
| + DAFA + FAWA | $82.93 \pm 0.17$ | $67.72 \pm 0.54$ | -0.01 | $50.06 \pm 0.11$ | $\mathbf{30.48 \pm 0.39}$ | $\mathbf{0.52}$ |

(b) FRL

| Method | Clean | | | Robust | | |
|---|---|---|---|---|---|---|
| | Average | Worst | $\rho_{nat}$ | Average | Worst | $\rho_{rob}$ |
| TRADES | $81.90 \pm 0.17$ | $65.20 \pm 0.66$ | - | $48.80 \pm 0.11$ | $20.50 \pm 1.49$ | - |
| + FRL | $\mathbf{82.97 \pm 0.30}$ | $\mathbf{71.94 \pm 0.97}$ | $\mathbf{0.12}$ | $44.68 \pm 0.27$ | $25.84 \pm 1.09$ | 0.18 |
| + DAFA | $80.85 \pm 0.44$ | $63.18 \pm 1.63$ | -0.04 | $\mathbf{48.02 \pm 0.03}$ | $\mathbf{29.14 \pm 0.55}$ | $\mathbf{0.41}$ |

(c) BAT

| Method | Clean | | | Robust | | |
|---|---|---|---|---|---|---|
| | Average | Worst | $\rho_{nat}$ | Average | Worst | $\rho_{rob}$ |
| TRADES | $83.50 \pm 0.24$ | $66.22 \pm 1.44$ | - | $49.53 \pm 0.26$ | $20.93 \pm 1.09$ | - |
| + BAT | $\mathbf{87.16 \pm 0.07}$ | $\mathbf{74.02 \pm 1.38}$ | $\mathbf{0.16}$ | $45.89 \pm 0.20$ | $17.82 \pm 1.09$ | -0.22 |
| + DAFA | $82.65 \pm 0.25$ | $67.94 \pm 1.54$ | 0.02 | $\mathbf{48.59 \pm 0.22}$ | $\mathbf{29.64 \pm 1.25}$ | $\mathbf{0.40}$ |

(d) WAT

| Method | Clean | | | Robust | | |
|---|---|---|---|---|---|---|
| | Average | Worst | $\rho_{nat}$ | Average | Worst | $\rho_{rob}$ |
| TRADES | $82.30 \pm 0.21$ | $65.62 \pm 1.22$ | - | $48.87 \pm 0.20$ | $21.52 \pm 1.09$ | - |
| + WAT | $80.97 \pm 0.29$ | $\mathbf{70.57 \pm 1.54}$ | $\mathbf{0.06}$ | $46.30 \pm 0.27$ | $28.55 \pm 1.30$ | 0.27 |
| + DAFA | $\mathbf{82.31 \pm 0.32}$ | $68.11 \pm 1.05$ | 0.04 | $\mathbf{49.03 \pm 0.20}$ | $\mathbf{29.93 \pm 1.38}$ | $\mathbf{0.39}$ |

Table 12: Comparison of the performance of the models for improving robust fairness methods in the setting of each method on the CIFAR-100 dataset: CFA (Wei et al., 2023), FRL (Xu et al., 2021), BAT (Sun et al., 2023), and WAT (Li & Liu, 2023). The best results among robust fairness methods are indicated in bold.

(a) CFA

| Method | Clean | | | Robust | | |
|---|---|---|---|---|---|---|
| | Average | Worst | $\rho_{nat}$ | Average | Worst | $\rho_{rob}$ |
| TRADES | $56.50 \pm 0.34$ | $18.20 \pm 1.56$ | - | $24.67 \pm 0.13$ | $1.07 \pm 0.85$ | - |
| + EMA | $59.14 \pm 0.13$ | $17.07 \pm 0.39$ | -0.02 | $26.80 \pm 0.12$ | $1.73 \pm 0.93$ | 0.71 |
| + CCM + CCR | $57.25 \pm 0.69$ | $19.60 \pm 1.97$ | 0.09 | $24.18 \pm 0.56$ | $1.20 \pm 0.75$ | 0.10 |
| + CCM + CCR + EMA | $\mathbf{59.50 \pm 0.17}$ | $\mathbf{22.40 \pm 1.29}$ | **0.28** | $\mathbf{26.48 \pm 0.13}$ | $1.53 \pm 0.09$ | 0.64 |
| + CCM + CCR + FAWA | $57.74 \pm 1.31$ | $15.00 \pm 1.55$ | -0.15 | $23.49 \pm 0.30$ | $1.67 \pm 1.25$ | 0.51 |
| + DAFA | $56.13 \pm 0.21$ | $17.20 \pm 1.56$ | -0.06 | $24.25 \pm 0.17$ | $1.87 \pm 0.62$ | 0.73 |
| + DAFA + EMA | $58.74 \pm 0.19$ | $17.07 \pm 1.06$ | -0.02 | $26.42 \pm 0.10$ | $\mathbf{2.20 \pm 0.65}$ | **1.13** |

(b) FRL

| Method | Clean | | | Robust | | |
|---|---|---|---|---|---|---|
| | Average | Worst | $\rho_{nat}$ | Average | Worst | $\rho_{rob}$ |
| TRADES | $57.47 \pm 0.15$ | $19.00 \pm 1.67$ | - | $25.50 \pm 0.11$ | $2.00 \pm 0.40$ | - |
| + FRL | $\mathbf{57.59 \pm 0.34}$ | $18.73 \pm 1.84$ | -0.01 | $24.93 \pm 0.29$ | $1.80 \pm 0.54$ | -0.12 |
| + DAFA | $57.07 \pm 0.18$ | $\mathbf{20.20 \pm 1.25}$ | **0.06** | $\mathbf{24.98 \pm 0.11}$ | $\mathbf{2.60 \pm 0.49}$ | **0.28** |

(c) BAT

| Method | Clean | | | Robust | | |
|---|---|---|---|---|---|---|
| | Average | Worst | $\rho_{nat}$ | Average | Worst | $\rho_{rob}$ |
| TRADES | $56.51 \pm 0.27$ | $16.40 \pm 0.49$ | - | $24.98 \pm 0.09$ | $1.00 \pm 0.00$ | - |
| + BAT | $\mathbf{62.80 \pm 0.17}$ | $\mathbf{22.40 \pm 2.15}$ | **0.48** | $23.42 \pm 0.08$ | $0.20 \pm 0.40$ | -0.86 |
| + DAFA | $56.64 \pm 0.35$ | $20.20 \pm 2.48$ | 0.23 | $\mathbf{24.16 \pm 0.17}$ | $\mathbf{2.00 \pm 0.63}$ | **0.97** |

(d) WAT

| Method | Clean | | | Robust | | |
|---|---|---|---|---|---|---|
| | Average | Worst | $\rho_{nat}$ | Average | Worst | $\rho_{rob}$ |
| TRADES | $57.96 \pm 0.06$ | $20.40 \pm 1.20$ | - | $25.41 \pm 0.08$ | $1.50 \pm 0.50$ | - |
| + WAT | $53.56 \pm 0.43$ | $17.07 \pm 2.29$ | -0.24 | $22.65 \pm 0.28$ | $1.87 \pm 0.19$ | 0.14 |
| + DAFA | $\mathbf{58.28 \pm 0.18}$ | $\mathbf{21.30 \pm 1.27}$ | **0.05** | $\mathbf{24.70 \pm 0.12}$ | $\mathbf{2.30 \pm 0.78}$ | **0.51** |

Table 13: Comparison of the performance of the models for improving robust fairness methods in the setting of each method on the STL-10 dataset: CFA (Wei et al., 2023), FRL (Xu et al., 2021), BAT (Sun et al., 2023), and WAT (Li & Liu, 2023). The best results among robust fairness methods are indicated in bold.

(a) CFA

| Method | Clean | | | Robust | | |
|---|---|---|---|---|---|---|
| | Average | Worst | $\rho_{nat}$ | Average | Worst | $\rho_{rob}$ |
| TRADES | $61.95 \pm 0.31$ | $38.62 \pm 1.22$ | - | $28.36 \pm 0.95$ | $5.51 \pm 0.51$ | - |
| + EMA | $62.28 \pm 0.25$ | $39.43 \pm 0.56$ | 0.03 | $31.75 \pm 0.86$ | $6.97 \pm 0.68$ | 0.38 |
| + CCM + CCR | $60.74 \pm 0.42$ | $37.02 \pm 1.69$ | -0.06 | $31.24 \pm 0.27$ | $6.82 \pm 0.65$ | 0.34 |
| + CCM + CCR + EMA | $60.75 \pm 0.54$ | $38.53 \pm 0.97$ | -0.02 | $31.83 \pm 0.14$ | $7.63 \pm 0.23$ | 0.51 |
| + CCM + CCR + FAWA | $60.70 \pm 0.57$ | $38.53 \pm 0.95$ | -0.02 | $\mathbf{31.88 \pm 0.14}$ | $7.69 \pm 0.27$ | 0.52 |
| + DAFA | $61.37 \pm 1.25$ | $43.29 \pm 1.46$ | 0.11 | $28.07 \pm 1.00$ | $7.93 \pm 0.59$ | 0.43 |
| + DAFA + EMA | $\mathbf{61.10 \pm 1.02}$ | $\mathbf{42.84 \pm 1.48}$ | $\mathbf{0.10}$ | $30.52 \pm 0.36$ | $\mathbf{11.02 \pm 0.55}$ | $\mathbf{1.08}$ |

(b) FRL

| Method | Clean | | | Robust | | |
|---|---|---|---|---|---|---|
| | Average | Worst | $\rho_{nat}$ | Average | Worst | $\rho_{rob}$ |
| TRADES | $61.52 \pm 0.27$ | $39.28 \pm 1.47$ | - | $30.68 \pm 0.21$ | $6.95 \pm 0.45$ | - |
| + FRL | $\mathbf{56.75 \pm 0.37}$ | $\mathbf{31.26 \pm 1.41}$ | $\mathbf{-0.28}$ | $28.99 \pm 0.39$ | $7.94 \pm 0.49$ | 0.09 |
| + DAFA | $61.00 \pm 0.20$ | $44.29 \pm 1.72$ | 0.12 | $\mathbf{29.77 \pm 0.15}$ | $\mathbf{9.24 \pm 0.53}$ | $\mathbf{0.30}$ |

(c) BAT

| Method | Clean | | | Robust | | |
|---|---|---|---|---|---|---|
| | Average | Worst | $\rho_{nat}$ | Average | Worst | $\rho_{rob}$ |
| TRADES | $62.13 \pm 0.27$ | $38.70 \pm 0.98$ | - | $30.07 \pm 0.28$ | $5.95 \pm 0.41$ | - |
| + BAT | $\mathbf{59.06 \pm 1.29}$ | $\mathbf{36.75 \pm 2.44}$ | $\mathbf{-0.10}$ | $23.37 \pm 0.98$ | $3.22 \pm 1.00$ | -0.68 |
| + DAFA | $61.13 \pm 0.34$ | $43.55 \pm 1.72$ | 0.11 | $\mathbf{28.73 \pm 0.37}$ | $\mathbf{8.96 \pm 0.85}$ | $\mathbf{0.46}$ |

(d) WAT

| Method | Clean | | | Robust | | |
|---|---|---|---|---|---|---|
| | Average | Worst | $\rho_{nat}$ | Average | Worst | $\rho_{rob}$ |
| TRADES | $58.01 \pm 0.16$ | $36.45 \pm 1.15$ | - | $29.75 \pm 0.25$ | $5.13 \pm 0.47$ | - |
| + WAT | $52.92 \pm 1.80$ | $30.82 \pm 1.44$ | -0.24 | $26.98 \pm 0.25$ | $6.72 \pm 0.75$ | 0.22 |
| + DAFA | $\mathbf{57.14 \pm 0.53}$ | $\mathbf{39.61 \pm 1.66}$ | $\mathbf{0.07}$ | $\mathbf{27.92 \pm 0.30}$ | $\mathbf{10.35 \pm 0.68}$ | $\mathbf{0.96}$ |

Table 14: Performance of the models for improving robust fairness methods in the same setting (Pang et al., 2021). The best results among robust fairness methods are indicated in bold.

(a) CIFAR-10

| Method | Clean | | | Robust | | |
|---|---|---|---|---|---|---|
| | Average | Worst | $\rho_{nat}$ | Average | Worst | $\rho_{rob}$ |
| TRADES | $82.04 \pm 0.10$ | $65.70 \pm 1.32$ | 0.00 | $49.80 \pm 0.18$ | $21.31 \pm 1.19$ | 0.00 |
| TRADES + FRL | $80.72 \pm 0.46$ | $71.33 \pm 1.38$ | 0.06 | $47.62 \pm 0.63$ | $26.47 \pm 1.54$ | 0.19 |
| TRADES + CFA | $76.79 \pm 0.58$ | $59.98 \pm 0.99$ | -0.15 | $48.32 \pm 0.49$ | $27.04 \pm 0.44$ | 0.24 |
| TRADES + BAT | $\mathbf{86.68 \pm 0.25}$ | $\mathbf{72.04 \pm 1.88}$ | $\mathbf{0.15}$ | $45.94 \pm 0.74$ | $17.59 \pm 1.83$ | -0.19 |
| TRADES + WAT | $80.38 \pm 0.36$ | $68.88 \pm 2.40$ | 0.03 | $46.95 \pm 0.33$ | $28.84 \pm 1.34$ | 0.30 |
| TRADES + DAFA | $81.63 \pm 0.20$ | $67.90 \pm 1.25$ | 0.03 | $\mathbf{49.05 \pm 0.14}$ | $\mathbf{30.53 \pm 1.04}$ | $\mathbf{0.42}$ |

(b) CIFAR-100

| Method | Clean | | | Robust | | |
|---|---|---|---|---|---|---|
| | Average | Worst | $\rho_{nat}$ | Average | Worst | $\rho_{rob}$ |
| TRADES | $58.34 \pm 0.38$ | $16.80 \pm 1.05$ | 0.00 | $25.60 \pm 0.14$ | $1.33 \pm 0.94$ | 0.00 |
| TRADES + FRL | $56.43 \pm 0.44$ | $17.67 \pm 0.47$ | 0.02 | $24.44 \pm 0.16$ | $1.33 \pm 0.94$ | -0.04 |
| TRADES + CFA | $58.66 \pm 0.26$ | $20.13 \pm 1.59$ | 0.20 | $24.73 \pm 0.15$ | $1.07 \pm 0.77$ | -0.23 |
| TRADES + BAT | $\mathbf{66.14 \pm 0.18}$ | $\mathbf{27.93 \pm 1.61}$ | $\mathbf{0.80}$ | $22.79 \pm 0.21$ | $0.13 \pm 0.34$ | -1.01 |
| TRADES + WAT | $56.37 \pm 0.32$ | $23.27 \pm 1.91$ | 0.35 | $23.72 \pm 0.24$ | $1.93 \pm 0.44$ | 0.38 |
| TRADES + DAFA | $58.07 \pm 0.05$ | $18.67 \pm 0.47$ | 0.11 | $\mathbf{25.08 \pm 0.10}$ | $\mathbf{2.33 \pm 0.47}$ | $\mathbf{0.73}$ |

(c) STL-10

| Method | Clean | | | Robust | | |
|---|---|---|---|---|---|---|
| | Average | Worst | $\rho_{nat}$ | Average | Worst | $\rho_{rob}$ |
| TRADES | $61.13 \pm 0.57$ | $39.29 \pm 1.71$ | 0.00 | $31.36 \pm 0.51$ | $7.73 \pm 0.99$ | 0.00 |
| TRADES + FRL | $58.29 \pm 0.44$ | $40.50 \pm 2.38$ | -0.02 | $28.28 \pm 0.34$ | $8.48 \pm 0.64$ | 0.00 |
| TRADES + CFA | $59.59 \pm 0.60$ | $39.86 \pm 0.45$ | -0.01 | $\mathbf{31.36 \pm 0.22}$ | $7.93 \pm 0.21$ | 0.03 |
| TRADES + BAT | $59.94 \pm 0.30$ | $37.81 \pm 1.84$ | -0.06 | $24.02 \pm 0.26$ | $3.59 \pm 1.03$ | -0.77 |
| TRADES + WAT | $56.85 \pm 1.10$ | $35.08 \pm 3.45$ | -0.18 | $28.49 \pm 0.96$ | $7.89 \pm 1.64$ | -0.07 |
| TRADES + DAFA | $\mathbf{60.50 \pm 0.57}$ | $\mathbf{42.23 \pm 2.26}$ | $\mathbf{0.06}$ | $29.98 \pm 0.46$ | $\mathbf{10.73 \pm 1.31}$ | $\mathbf{0.34}$ |

