# OpenReview forum: "DAFA: Distance-Aware Fair Adversarial Training"
_ICLR.cc/2024/Conference — ICLR 2024 poster_

### Official Review · Reviewer_Wxgz · 2023-10-23

**Soundness:** 3 good
**Presentation:** 3 good
**Contribution:** 3 good
**Rating:** 6
**Confidence:** 4

**Summary:**

This research addresses the "robust fairness problem" in adversarial training, where there's a significant difference in model accuracy between classes. Current methods sacrifice performance on easier classes to improve harder ones. However, the study observes that the model's predictions for the worst class often favor similar classes instead of the easy ones under adversarial attacks. As the distance between classes decreases, robust fairness deteriorates. To mitigate this, the Distance-Aware Fair Adversarial training (DAFA) approach is introduced. It assigns distinct loss weights and adversarial margins to each class and adjusts them to balance robustness among similar classes. Experiments show that DAFA not only maintains average robust accuracy but also significantly enhances fairness, especially for the worst-performing class, compared to existing methods.

**Strengths:**

1. The paper is easy to follow

2. The motivation is clear

**Weaknesses:**

1. Experiments on the sensitivity of trade-off hyperparameter $\beta$ is missing.

2. The experiments only based on TRADES are very limited.

3.  Several baselines are missing. For example, Group-DRO [1]

[1] Sagawa, Shiori, et al. "Distributionally Robust Neural Networks." International Conference on Learning Representations. 2019.

**Questions:**

See Weakness

---

> ### Author Response · Authors · 2023-11-17
> **Response to Reviewer Wxgz (1/2)**
>
> Thank you for reviewing our manuscript and pointing out the deficiencies in our paper's experiments. We appreciate the opportunity to address your concerns and questions.
> \
> \
> **Q**: **Experiments on the sensitivity of trade-off hyperparameter beta is missing.**
> \
> **A**: In robust fairness research, sensitivity ablation studies for the hyperparameter $\beta$ are generally not conducted, and experiments are typically carried out with a common fixed value of $\beta$, usually 6.0. Therefore, we did not perform an ablation study for this parameter. However, considering that changes in $\beta$ may have an impact, we will conduct experiments in response to the reviewer's suggestion. By varying the values of $\beta$ (1.0, 3.0, 6.0, and 9.0) and applying our method, we compare the enhanced robust fairness with the baseline TRADES model. As shown in the results below, in all settings, we observe that the clean/average robust accuracy performance is maintained close to the baseline, while significantly improving the worst robust accuracy.
> |  Model | Clean Avg. | Clean Worst | $\rho_{nat}$ | Robust Avg. | Robust Worst | $\rho_{rob}$ |
> |:--------|:--------:|:--------:|:--------:|:--------:|:--------:|:--------:|
> |  TRADES ($\beta=1.0$) |87.778 $\pm$ 0.22| 74.700 $\pm$ 1.49 | - | 43.704 $\pm$ 0.29 | 16.000 $\pm$ 1.15 | - |
> |  TRADES ($\beta=1.0$) + DAFA (ours) | 87.430 $\pm$ 0.23 | 75.220 $\pm$ 1.15 | 0.00 | 43.120 $\pm$ 0.25 | 20.620 $\pm$ 1.14 | 0.28 |
> |  TRADES ($\beta=3.0$) |85.030 $\pm$ 0.19| 70.600 $\pm$ 1.57 | - | 48.644 $\pm$ 0.35 | 21.100 $\pm$ 1.46 | - |
> |  TRADES ($\beta=3.0$) + DAFA (ours) | 84.582 $\pm$ 0.14 | 71.340 $\pm$ 1.37 | 0.01 | 47.676 $\pm$ 0.16 | 27.440 $\pm$ 1.80 | 0.28 |
> |  TRADES ($\beta=6.0$) |82.040 $\pm$ 0.10 | 65.700 $\pm$ 1.32 | - | 49.800 $\pm$ 0.18 | 21.310 $\pm$ 1.19 | - |
> |  TRADES ($\beta=6.0$) + DAFA (ours) | 81.630 $\pm$ 0.08 | 67.900 $\pm$ 1.25 | 0.03 | 49.050 $\pm$ 0.14 | 30.530 $\pm$ 1.04 | 0.42 |
> |  TRADES ($\beta=9.0$) |79.944 $\pm$ 0.13 | 63.000 $\pm$ 1.65 | - | 49.850 $\pm$ 0.24 | 20.860 $\pm$ 1.38 | - |
> |  TRADES ($\beta=9.0$) + DAFA (ours) | 79.492$\pm$ 0.10 | 65.360 $\pm$ 2.07 | 0.03 | 48.864 $\pm$ 0.17 | 29.800 $\pm$ 1.54 | 0.41 |
> ||||||||||
>
> \
> **Q**: **The experiments only based on TRADES are very limited.**
> \
> **A**: In addition to the TRADES method, the results of applying our approach to PGD can be found in Appendix E (Pages 22-23). Generally, in research related to adversarial training, TRADES and PGD are two major learning algorithms commonly applied. We followed this convention in our study. However, as the impact of applying our method to other baseline algorithms may vary, we explore the application of our approach to another adversarial training algorithm, adversarial weight perturbation (AWP) [1], and the results are as follows. The experimental setup mirrors that of TRADES for ResNet18, and the results demonstrate that applying our method to AWP, like with the existing results, significantly improves robust worst accuracy.
>
> | Model | Clean Avg. | Clean Worst | $\rho_{nat}$ | Robust Avg. | Robust Worst | $\rho_{rob}$ |
> |:--------|:--------:|:--------:|:--------:|:--------:|:--------:|:--------:|
> |  AWP |81.325 $\pm$ 0.02| 64.300 $\pm$ 1.10 | - | 49.970 $\pm$ 0.18 | 20.650 $\pm$ 0.45 | - |
> |  AWP + DAFA (ours) | 80.120 $\pm$ 0.07 | 66.740 $\pm$ 0.35 | 0.02 | 49.416 $\pm$ 0.16 | 28.260 $\pm$ 1.25 | 0.36 |
> ||||||||||
>
> [1] Wu, Dongxian, Shu-Tao Xia, and Yisen Wang. "Adversarial weight perturbation helps robust generalization." Advances in Neural Information Processing Systems 33 (2020): 2958-2969.

---

> ### Author Response · Authors · 2023-11-17
> **Response to Reviewer Wxgz (2/2)**
>
> **Q**: **Several baselines are missing. For example, Group-DRO.**
> \
> **A**: We conducted a comparison with the four baseline methods (FRL, CFA, BAT, and WAT) in our paper for a specific reason. Recent papers proposing methods primarily showcase state-of-the-art performance, hence we conducted comparisons with these methods (FRL 2021, CFA 2023, BAT 2023, and WAT 2023). We appreciate the recommendation of a paper for baseline comparison. After reviewing the suggested paper, we find that the Group-DRO method deals with datasets structured as $(x, y, a)$ (data, label, attribute), grouping data with the same attribute into groups. In this scenario, it trains the model parameters through the empirical risk minimization (ERM) of the group with the highest loss within a specific attribute group $(x|y,a)$ in one class $(x|y)$. As a result, it enhances the performance of groups showing low performance within a class, improving class data performance, and reduces bias due to attributes by preventing the learning of spurious correlations, thus minimizing performance differences. In summary, Group-DRO aims to reduce performance differences between groups within a class. However, our robust fairness research, covered in this paper, aims to reduce performance differences between classes. Therefore, we use datasets such as CIFAR-10, CIFAR-100, and STL-10, which have only class label information without attributes.
> \
> \
> Applying the Group-DRO method, designed for datasets containing attributes, to the datasets (CIFAR-10, CIFAR-100, and STL-10) used in our paper poses challenges. Conversely, to apply our method to datasets used in Group-DRO (CelebA, Waterbird, and MultiNLI) that include attributes, we need to consider the following. First, the performance evaluation of the Group-DRO method measures the accuracy (Worst-**"Group"** Accuracy) of groups showing low performance. For a fair comparison, we need to measure the accuracy of our proposed method for a specific attribute group. However, our method, proposed to reduce performance differences between classes, evaluates performance by measuring accuracy for a class showing low performance (Worst-**"Class"** Accuracy). Aligning these objectives requires transforming our method into one that aims to reduce performance differences between groups rather than classes. Considering the lengthy training time for adversarial training and the change in the task itself, such adjustments extend beyond the scope of our research. Consequently, comparing Group-DRO and our proposed method is challenging for the reasons mentioned; therefore, we plan to explore it as future work.
> \
> \
> Instead, we include additional experiments comparing our method with another proposed method in the field of robust fairness, cost-sensitive learning (CSL) [2]. The table below presents the experimental results when applying CSL to the CIFAR-10 dataset in the settings used in our paper (ResNet18). Comparing the results with CSL, it is evident that our method continues to exhibit superior performance compared to existing methods.
> | Model | Clean Avg. | Clean Worst | $\rho_{nat}$ | Robust Avg. | Robust Worst | $\rho_{rob}$ |
> |:--------|:--------:|:--------:|:--------:|:--------:|:--------:|:--------:|
> |  TRADES |82.040 $\pm$ 0.10 | 65.700 $\pm$ 1.32 | - | 49.800 $\pm$ 0.18 | 21.310 $\pm$ 1.19 | - |
> |  TRADES + CSL|82.070 $\pm$ 0.13 | 66.750 $\pm$ 0.25 | 0.02 | 49.740 $\pm$ 0.14 | 22.850 $\pm$ 0.45 | 0.07 |
> |  TRADES + DAFA (ours) | 81.630 $\pm$ 0.08 | 67.900 $\pm$ 1.25 | 0.03 | 49.050 $\pm$ 0.14 | 30.530 $\pm$ 1.04 | 0.42 |
> ||||||||||
>
> [2] Benz, Philipp, et al. "Robustness may be at odds with fairness: An empirical study on class-wise accuracy." NeurIPS 2020 Workshop on Pre-registration in Machine Learning. PMLR, 2021.

---

> > ### Author Response · Authors · 2023-11-21
> >
> > Dear reviewer Wxgz
> >
> > Thank you again for your insightful comments and suggestions!  We are sincerely thankful for your effort in reviewing our manuscript. As a gentle reminder, since there are only very few days left, we sincerely look forward to your follow-up response.
> >
> > we have submitted a rebuttal and a revised manuscript to address your mentioned concerns. We are happy to provide additional answers to illustrate the strength of our paper. In our previous responses, we carefully read your comments and made detailed responses summarized below:
> >
> > - We conducted experiments on the hyperparameter $\beta$ of TRADES. The results show that, despite changes in the setting depending on the value of $\beta$, it remains effective in improving robust fairness.
> > - We performed experiments applying our methodology to other adversarial training algorithms. In addition to applying DAFA to previously reported TRADES and PGD, we further demonstrated the utility of our approach by applying it to AWP, improving robust fairness. Through this, we showed that our methodology can be applied to various baseline algorithms.
> > - We thoroughly examined the previous study recommended by the reviewer (Group-DRO) and concluded that comparing Group-DRO and our proposed method is challenging. Instead, we conducted further experiments comparing with other existing methodologies (CSL). The comparison results demonstrated that our methodology is still superior to traditional methodologies.
> >
> > We hope that the provided new experiments and the additional discussion have convinced you of the merits of this paper. Please do not hesitate to contact us if there are additional questions.
> >
> > Meanwhile, we would like to thank the reviewer again for the very helpful comments. By taking them into account, it would indeed make our paper clearer and stronger.
> >
> > Thank you for your time and effort!
> >
> > Best regards, \
> > Authors

---

### Official Review · Reviewer_73vn · 2023-10-30

**Soundness:** 3 good
**Presentation:** 3 good
**Contribution:** 3 good
**Rating:** 8
**Confidence:** 4

**Summary:**

This paper aims to mitigate the class-wise fairness issue in adversarial training (AT) by proposing a DAFA framework, which first theoretically and empirically investigates the impact of the distance between classes on worst-class robustness and then proposes dynamically adjusting the weights and margins deployed in AT for better robust fairness.

**Strengths:**

1. The motivation of this paper is clear. Figure 1-3 illustrates what similar and dissimilar classes are and how they affect class-wise performance.
2. The claims of the influence of class-wise distance on their robustness are supported by both theoretical analysis and empirical verification.
3. This paper further refines the state-of-the-art understanding of class-wise accuracy and robustness. As CFA [4] shows, different classes should use specific training configurations in AT to strengthen class-wise performance, the proposed DAFA further takes the interaction of classes into consideration and highlights the role of similarity between classes.

**Weaknesses:**

1. My major concern with this work is the unfair comparison with baselines in terms of experiment settings. Here, I take CFA [4] as an example, and please also check other baselines.
    - CFA includes a Fairness-Aware Weight-Average (FAWA) strategy to mitigate the fluctuating effect of worst-class robustness. Weight average requires more epochs to converge after learning rate decay, and in the original paper of CFA, the learning rate decays at the 100th and 150th epoch in a 200-epoch training. Therefore, the learning rate schedule in your setting significantly decreases the performance of CFA.

    I suggest using the original settings of different baselines to ensure fair comparisons. I will change my rating based on how this issue is addressed.

2. Page 1. The research thread of the robust fairness problem seems to be confusing. Based on my expertise in robust fairness research, the main thread [1-5] on this problem focuses more on how to adjust objective functions and configurations in AT, and the long-tailed classification is a minor perspective. I suggest revising the introduction of this paper by focusing on more related work [1-5].
3. Page 4. Some of the Theorems seem to be very close to previous work [1, 3]. I suggest clarifying which theorems are a re-deduction of previous results and which are proposed new ones.
4. There may be a mismatch between the motivation and the proposed method. It seems that the schedule in equation (6)-(9) still pays more attention to easy or hard, though class-wise similarity is involved. I suggest adding theoretical analysis on how these configurations improve class-wise robustness.

### Minor Comments


- Page 2: Typo
> limiting the learning of these neighboring classes can serve as an effective means to improve robust fairness.
- Page 2, Section 2: I suggest adding the optimization objective of adversarial training (AT), which is a more primary method than TRADES.
- Page 3, Section 3: It may benefit from introducing what variance is in this context more detailedly and formally.
- Page 3, Section 3: Misuse of "On the other hand". "In addition" is suggested.
- Page 4. I suggest putting the details on how to calculate class-wise distance in Section 3 rather than the appendix to improve readability.
- Page 6, Section 4:
> we employ two strategies: adjusting loss weights and adversarial margins

    is very similar to the CCM and CCR strategies proposed in [4]. A reference is required here.


[1] To be robust or to be fair: Towards fairness in adversarial training. ICML

[2] Analysis and applications of class-wise robustness in adversarial training.KDD

[3] On the tradeoff between robustness and fairness. NeurIPS

[4] CFA: Class-wise calibrated fair adversarial training. CVPR

[5] WAT: improve the worst-class robustness in adversarial training. AAAI

**Questions:**

1. Page 5, Section 3. Why is class-wise distance defined as the average distance of a class to other classes? In my opinion, for classification tasks, the distance between a class and its closest one plays the most important role in determining the decision boundary, and the influences of classes that are far away are negligible.

---

> ### Author Response · Authors · 2023-11-17
> **Response to Reviewer 73vn (1/4)**
>
> Thank you for reviewing our manuscript, highlighting certain aspects in the experimental results that raised questions, and providing guidance on major and minor corrections.
> \
> \
> **Q**: **My major concern with this work is the unfair comparison with baselines in terms of experiment settings. Here, I take CFA as an example, and please also check other baselines.**
> \
> **A**: In response to the reviewer's feedback, we revisited our method's comparison within the settings of the methods we examined. Consequently, across all settings, our results consistently demonstrate effects that align with the outcomes presented in the paper, providing evidence of improving robust fairness.
> \
> \
> Before delving into the comparative results, it's essential to note that the existing methods compared in our paper were proposed with distinct LR scheduling strategies. FRL involves 40 epochs of fine-tuning after pretraining, WAT utilizes the 75-90-100 LR scheduling identical to the TRADES paper [1], and both CFA and BAT adopt the 100-150-200 LR scheduling following the PGD paper [2]. The reason for employing a uniform 100-105-110 LR scheduling in our paper for all methods is rooted in the findings of the bag-of-tricks paper [3], which suggests that the TRADES model demonstrates the highest robustness performance under such LR scheduling, without exhibiting significant robust overfitting [4]. This LR scheduling is chosen to evaluate the effectiveness of robust fairness in settings where robust overfitting is not prominently observed.
> \
> \
> The LR scheduling of 100-150-200 is commonly used in the research domain that addresses robust overfitting since it tends to occur in such settings. It's important to note that among the robust fairness methods, only CFA has the capability to address robust overfitting, incorporating FAWA, a variant of EMA. However, it's crucial to highlight that our method, along with other existing robust fairness methods, does not explicitly address robust overfitting. Therefore, if experiments were conducted using the 100-150-200 LR scheduling, which is advantageous for CFA in environments where robust overfitting occurs, a fair comparison would require incorporating robust overfitting methods (e.g., EMA) alongside the compared methods. CFA's paper may consider this aspect by incorporating EMA into the baseline algorithm for a fair comparison. Considering all these factors, we opted for a unified comparison under the LR scheduling proposed by the bag-of-tricks paper.
> \
> \
> Taking into consideration the variations in experimental settings for each method and considering the reviewer's comments, we conducted experiments for our proposed method within the specific settings of each method.
> \
> \
> [1] Zhang, Hongyang, et al. "Theoretically principled trade-off between robustness and accuracy." International conference on machine learning. PMLR, 2019. \
> [2] Madry, Aleksander, et al. "Towards Deep Learning Models Resistant to Adversarial Attacks." International Conference on Learning Representations. 2018. \
> [3] Pang, Tianyu, et al. "Bag of Tricks for Adversarial Training." International Conference on Learning Representations. 2020. \
> [4] Rice, Leslie, Eric Wong, and Zico Kolter. "Overfitting in adversarially robust deep learning." International Conference on Machine Learning. PMLR, 2020.
>
> \
> **CFA**
> \
> In consideration of the aforementioned concerns related to robust overfitting, we compare the results when EMA is employed. The model is trained using a 100-150-200 LR scheduling with the architecture being PreAct ResNet18. Our method's settings align with those presented in the paper. The baseline model incorporates EMA in the same way as CFA, and our method applies EMA after 20 epochs of training following the method's application. For all models, we measure the average performance of the five checkpoints after the second LR decay (151~155).
> |  Model | Clean Avg. | Clean Worst | $\rho_{nat}$ | Robust Avg. | Robust Worst | $\rho_{rob}$ |
> |:--------|:--------:|:--------:|:--------:|:--------:|:--------:|:--------:|
> |  TRADES |84.080 $\pm$ 0.14| 67.220 $\pm$ 2.22 | - | 48.914 $\pm$ 0.18 | 20.420 $\pm$ 1.69 | - |
> |  TRADES + EMA |83.966 $\pm$ 0.15| 66.900 $\pm$ 0.17 | -0.01 | 50.620 $\pm$ 0.10 | 22.160 $\pm$ 0.41 | 0.12 |
> |  TRADES + CCM + CCR | 81.722 $\pm$ 0.26| 65.840 $\pm$ 2.07 | -0.05 | 49.818 $\pm$ 0.22 | 22.580 $\pm$ 1.72 | 0.12 |
> |  TRADES + CCM + CCR + EMA| 81.680 $\pm$ 0.17| 66.200 $\pm$ 0.22 | -0.04 | 51.010 $\pm$ 0.06 | 23.160 $\pm$ 0.10 | 0.18 |
> |  TRADES + CCM + CCR + FAWA| 79.806 $\pm$ 0.15| 65.180 $\pm$ 0.16 | -0.08 | 50.780 $\pm$ 0.05 | 27.620 $\pm$ 0.04 | 0.39 |
> |  TRADES +DAFA (ours) | 82.845 $\pm$ 0.05 | 68.475 $\pm$ 0.68 | 0.00 | 48.360 $\pm$ 0.18 | 29.680 $\pm$ 0.77 | 0.44 |
> |  TRADES +DAFA (ours) +EMA | 83.190 $\pm$ 0.02 | 70.400 $\pm$ 0.07 | 0.04 | 50.632 $\pm$ 0.03 | 29.950 $\pm$ 0.11 | 0.50 |
> ||||||||||

---

> ### Author Response · Authors · 2023-11-17
> **Response to Reviewer 73vn (2/4)**
>
> **FRL**
> \
> Using the pretrained model provided by the official code GitHub (https://github.com/hannxu123/fair_robust), we conduct model training with 40 epochs of fine-tuning using PreAct ResNet18 architecture. Due to the low performance of the pretrained model, after 10 epochs of fine-tuning, we compute class weight $\mathcal{W}$ and apply our method for the remaining 30 epochs. For all models, we measure the average performance of the last five checkpoints (36~40).
> |  Model | Clean Avg. | Clean Worst | $\rho_{nat}$ | Robust Avg. | Robust Worst | $\rho_{rob}$ |
> |:--------|:--------:|:--------:|:--------:|:--------:|:--------:|:--------:|
> |  TRADES |81.900 $\pm$ 0.17| 65.200 $\pm$ 0.66 | - | 48.798 $\pm$ 0.11 | 20.500 $\pm$ 1.49 | - |
> |  TRADES + FRL |82.966 $\pm$ 0.30| 71.940 $\pm$ 0.97 | 0.12 | 44.682 $\pm$ 0.27 | 25.840 $\pm$ 1.09 | 0.18 |
> |  TRADES + DAFA (ours) | 80.848 $\pm$ 0.44| 63.180 $\pm$ 1.63 | -0.04 | 48.018 $\pm$ 0.03 | 29.140 $\pm$ 0.55 | 0.41 |
> ||||||||||
>
> \
> **BAT**
> \
> The model is trained with a 100-150-200 LR scheduling, and the architecture is ResNet18. For all models, we measure the average performance of the five checkpoints after the second LR decay (151~155).
> |  Model | Clean Avg. | Clean Worst | $\rho_{nat}$ | Robust Avg. | Robust Worst | $\rho_{rob}$ |
> |:--------|:--------:|:--------:|:--------:|:--------:|:--------:|:--------:|
> |  TRADES |83.495 $\pm$ 0.24| 66.220 $\pm$ 1.44 | - | 49.531 $\pm$ 0.26 | 20.927 $\pm$ 1.09 | - |
> |  TRADES + BAT | 87.160 $\pm$ 0.07 | 74.020 $\pm$ 1.38 | 0.16 | 45.894 $\pm$ 0.20 | 17.820 $\pm$ 1.09 | -0.22 |
> |  TRADES + DAFA (ours) | 82.650 $\pm$ 0.25| 67.940 $\pm$ 1.54 | 0.02 | 48.594 $\pm$ 0.22 | 29.640 $\pm$ 1.25 | 0.40 |
> ||||||||||
>
> \
> **WAT**
> \
> The model is trained with a 75-90-100 LR scheduling, and the architecture is ResNet18. For all models, we measure the average performance of the five checkpoints after the second LR decay (91~95).
> |  Model | Clean Avg. | Clean Worst | $\rho_{nat}$ | Robust Avg. | Robust Worst | $\rho_{rob}$ |
> |:--------|:--------:|:--------:|:--------:|:--------:|:--------:|:--------:|
> |  TRADES |82.299 $\pm$ 0.21| 65.620 $\pm$ 1.22 | - | 48.865 $\pm$ 0.20 | 21.520 $\pm$ 1.09 | - |
> |  TRADES + WAT | 80.966 $\pm$ 0.29 | 70.573 $\pm$ 1.54 | 0.06 | 46.301 $\pm$ 0.27 | 28.547 $\pm$ 1.30 | 0.27 |
> |  TRADES + DAFA (ours) | 82.312 $\pm$ 0.32| 68.113 $\pm$ 1.05 | 0.04 | 49.033 $\pm$ 0.20 | 29.933 $\pm$ 1.38 | 0.39 |
> ||||||||||
>
> Examining the results, in all settings, we observe that our method demonstrates average robust accuracy and clean accuracy comparable to the baseline algorithm. Simultaneously, it demonstrates a high level of worst robust accuracy.
> \
> \
> **Q**: **Page 1. The research thread of the robust fairness problem seems to be confusing. Based on my expertise in robust fairness research, the main thread on this problem focuses more on how to adjust objective functions and configurations in AT, and the long-tailed classification is a minor perspective. I suggest revising the introduction of this paper by focusing on more related work.**
> \
> **A**: Our objective was to present the challenges of robust fairness within the context of the more familiar subject of long-tailed classification. This choice was made to enhance accessibility for those unfamiliar with robust fairness. Nonetheless, if this approach is deemed inappropriate for introducing the paper's subject, we are open to alternative strategies. As advised, one option under consideration is to incorporate references to existing papers and revise the introduction section. This revision could involve illustrating problems that may arise with the deterioration of robust fairness, including scenarios where the worst class accuracy is exceptionally low. Since modifying the opening of the paper is a crucial task, we will carefully contemplate and make the necessary revisions.

---

> ### Author Response · Authors · 2023-11-17
> **Response to Reviewer 73vn (3/4)**
>
> **Q**: **Page 4. Some of the Theorems seem to be very close to previous work. I suggest clarifying which theorems are a re-deduction of previous results and which are proposed new ones.**
> \
> **A**: The theoretical model utilized in our theoretical analysis can be regarded as a more comprehensive version, encompassing the theoretical models mentioned by the reviewer in previous studies and integrating the concept of distance between two classes. In Eq. 2, which represents the distribution addressed in the theoretical model, when $\alpha=1$, it corresponds to models explored in earlier studies. Consequently, there might be a re-deduction of the proof formula from previous papers during the proof process.
> \
> \
> However, concerning the theorems presented in the main paper, it is ambiguous to term the theorems a re-deduction since they clarify the influence of changes in the distance between two classes in the newly augmented theoretical model on standard error and robust error. The proof of the theory does involve a re-deduction. Given that Lemma 1 is a re-deduction from a previous paper, we have acknowledged it. The proof of Theorem 1 can be considered a re-deduction up to the midpoint, as it incorporates an altered version of the error function addressed in the previous studies, which includes the novel concept of distance represented by $\alpha$. Additionally, the subsequent proof constitutes the new ones. Considering these aspects and incorporating the reviewer's feedback, we clarify the re-deduction section in the proof.
> \
> \
> **Q**: **There may be a mismatch between the motivation and the proposed method. It seems that the schedule in equation (6)-(9) still pays more attention to easy or hard, though class-wise similarity is involved. I suggest adding theoretical analysis on how these configurations improve class-wise robustness.**
> \
> **A**: As the reviewer correctly understood, our method utilizes the difficulty of classes. The methods for measuring class difficulty, as mentioned in the paper, can broadly be categorized into two: variance and distance. Previous works in robust fairness research have conducted theoretical analyses using variance. In contrast, we pursue a novel approach, conducting theoretical and empirical analyses using distance, specifically class-wise similarity or inter-class distance. Although we experimentally asserted that inter-class distance better reflects class difficulty than variance, we did not render variance meaningless or deny its impact. Variance still proves effective in binary classification task settings, playing a crucial role in comparing the relative difficulty of two classes. We considered these factors in various analytical experiments such as analysis in Figs. 3 and 5.
> \
> \
> In summary, our method leverages class difficulty through a new approach, either class-wise similarity or inter-class distance. As the reviewer noted, there may be doubts about the theoretical reason of how our proposed method improves class-wise robustness based on these new approaches. We have carefully considered this, recognizing the multitude of factors, such as task settings necessary for theoretical analysis, characteristics of classes in new settings, and relationships between classes. However, due to the difficulty of incorporating these considerations into a concrete theoretical analysis within a short timeframe, we will continue attempting to add this in the final version.
> \
> \
> **Q**:**Minor comments**
> \
> **A**: Thank you for thoroughly reviewing our paper. We appreciate your feedback on various aspects, including typos, preliminary content, correction guidance, readability, and references. We will address all the mentioned points in the revised version, considering page limits.
>
> - **Q**: **Page 2: Typo: limiting the learning of these neighboring classes can serve as an effective means to improve robust fairness.**
> \
> **A**:  We couldn't find typos in this sentence. Are you referring to semantic errors?
> - **Q**: **Page 3, Section 3: Misuse of "On the other hand". "In addition" is suggested.**
> \
> **A**: To contrast variance and inter-class distance, we have employed the phrase "On the other hand."
> - **Q**: **Page 2, Section 2: I suggest adding the optimization objective of adversarial training (AT), which is a more primary method than TRADES.** \
> **Q**: **Page 3, Section 3: It may benefit from introducing what variance is in this context more detailedly and formally.** \
> **Q**: **Page 4. I suggest putting the details on how to calculate class-wise distance in Section 3 rather than the appendix to improve readability.**
> \
> **A**: Insufficient contents have been either excluded or moved to the Appendix due to page constraints. In the revised version, we will seek ways to supplement these details by ensuring additional space.
>
> \
> (Continued)

---

> ### Author Response · Authors · 2023-11-17
> **Response to Reviewer 73vn (4/4)**
>
> - **Q**: **Page 6, Section 4: we employ two strategies: adjusting loss weights and adversarial margins** \
> **A**: Regarding the references to the method, the approach of utilizing different class weights for each class to enhance robust fairness is a common strategy employed in many existing methods, including the CFA [5] method that proposed using different class weights for each class to improve robust fairness. The study introducing FRL [6] proposed a method applying different class weights to cross-entropy loss, boundary loss, and margin for each class for the first time. The study introducing CFA [5] also applies different class weights to cross-entropy loss, boundary loss, and margin. Similarly, WAT [7] applies different class weights to cross-entropy loss and boundary loss. Our method applies class weights to cross-entropy loss and margin. Since the application of different class weights aligns with the approach adopted by most existing methods, we did not specifically cite a particular previous study. However, although the method for determining class weights may differ, because we follow the flow of applying different class weights in line with existing methods (FRL, CFA, and WAT), we will add references to these existing methods in the revised version. Due to page constraints, adjustments to the overall content are necessary to secure space in the main text. After making space-saving measures, the following sentence will be inserted after the sentence mentioned by the reviewer: "Adjusting loss weights or adversarial margins has been a common practice in previous methods [FRL, CFA, WAT] for enhancing robust fairness. In this context, we specifically delve into the adjustment of the decision boundary (DB) for the hard class."
>
> [5] Wei, Zeming, et al. "Cfa: Class-wise calibrated fair adversarial training." Proceedings of the IEEE/CVF Conference on Computer Vision and Pattern Recognition. 2023. \
> [6] Xu, Han, et al. "To be robust or to be fair: Towards fairness in adversarial training." International conference on machine learning. PMLR, 2021. \
> [7] Li, Boqi, and Weiwei Liu. "WAT: improve the worst-class robustness in adversarial training." Proceedings of the AAAI Conference on Artificial Intelligence. Vol. 37. No. 12. 2023.
> \
> \
> **Q**: **Page 5, Section 3. Why is class-wise distance defined as the average distance of a class to other classes? In my opinion, for classification tasks, the distance between a class and its closest one plays the most important role in determining the decision boundary, and the influences of classes that are far away are negligible.**
> \
> **A**: The message we intended to convey through Fig. 4 is that, in previous papers, class difficulty was theoretically modeled through variance and analyzed accordingly. However, the methods expressed class difficulty through accuracy. In reality, we observed a significant correlation between distance and accuracy. As mentioned, min distance can also be employed to convey this message. However, from the perspective of reflecting class difficulty, it is challenging to express it using min distance.
> \
> \
> For example, consider a 3-class classification with distances between two classes being 1, 2, and 3. If average distance is used, the average distances are 1.5, 2, and 2.5, leading to distinguishable calculations of difficulty for each class. However, using min distance, the two classes become 1, with one class being 2, making it impossible to compare the relative difficulty between the two classes representing the minimum distance. In the below Table, we calculate class-wise average distance, min distance, and accuracy. As seen in the results, occurrences of the minimum distance being the same between two different classes are frequent.
>
> |  Class | Plane | Car | Bird | Cat | Deer | Dog | Horse | Frog | Ship | Truck |
> |:--------:|:--------:|:--------:|:--------:|:--------:|:--------:|:--------:|:--------:|:--------:|:--------:|:--------:|
> | Average distance |2.663|	2.810|	2.063|	1.996|	2.118|	2.168|	2.275|	2.396|	2.616|	2.609|
> | Min distance | 2.110|	2.173|	1.471|	1.121|	1.471|	1.121|	1.614|	2.019|	2.110|	2.173|
> | Accuracy | 60.27|	72.22	|31.32	|21.31	|28.89|	40.65|	50.72|	59.73|	67.08|	65.84|
> ||||||||||||
>
>
> Another issue observable is the case where only one class is close, as exemplified by the dog class in the above. From the min distance perspective, cat and dog share the same class difficulty, but their actual performance shows nearly a twofold difference. This is due to the average distance from other classes, excluding each other, being relatively closer to cat than to dog. Additionally, the notion of "far away" in the question is ambiguous. There might be instances where the difference between the closest class and the second, third, fourth, and so on is quite small. Therefore, instead of adopting min distance considering only the closest class, we utilized average distance.

---

> ### Comment · Reviewer_73vn · 2023-11-17
> **Can you updated the experimental results in Table 1?**
>
> Dear authors,
>
> Thanks for your detailed and prompt reply. Most of my concerns are adequately addressed. However, I noticed that the main comparison in Table 1 remained unchanged. As expected, the performance of baselines with original settings is improved, and the gap between them and DATA is notably decreased. While I'm not criticizing the contribution of this paper, I suggest incorporating these results in your main comparison to ensure a more fair comparison.

---

> ### Author Response · Authors · 2023-11-17
> **Response to Reviewer 73vn "Can you updated the experimental results in Table 1?"**
>
> Dear reviewer 73vn
> \
> \
> Thank you for your prompt response and the valuable suggestions. As the reviewer suggested, we replace the comparison experiment results above with those in Table 1 of the main paper. Additionally, we revise the experimental explanations in both the main paper and Appendix D. We compile the experiment results as tables in Appendix E (Page 25). As observed in the results, our methodology exhibits comparable or superior clean/robust average accuracy to other methodologies in the same settings while demonstrating superior robust worst accuracy across all settings.
> Additionally, as explained in Response to Reviewer n9NT (2/2), we wish to highlight that our method incurs an extra time cost of less than 1 second throughout the entire training process for a 10-class dataset. Therefore, our methodology can be considered a cost-effective approach that enhances performance compared to existing methodologies. Regarding the results of experiments on other datasets, due to time constraints, we conducted experiments only on CIFAR-10. Although we are currently conducting experiments on CIFAR-100 and STL-10, it might be challenging to present the results within the given timeframe. We plan to include these results in the final version, both in the main paper and the appendix.

---

> ### Comment · Reviewer_73vn · 2023-11-18
> **Thank you for your efforts**
>
> Dear authors,
>
> Thanks for the update, I have raised my rating. There is also an optional experiment: given the significance and scalability of FAWA proposed in the CFA paper, can DATA+FAWA further improve robust fairness? Last but not least, please remember to incorporate updated experiments on other datasets in your final revision.

---

> > ### Author Response · Authors · 2023-11-18
> > **Response to Reviewer 73vn**
> >
> > Dear reviewer 73vn,
> > \
> > \
> > Thanks for your response. As the reviewer mentioned, incorporating FAWA into our methodology is feasible, and we anticipate that this integration could further enhance the performance of our approach. We appreciate your suggestion and plan to conduct experiments to verify this. Additionally, we will update the results in the main paper with completed experiments on other datasets, and move the existing results to the appendix.

---

### Official Review · Reviewer_n9NT · 2023-10-30

**Soundness:** 3 good
**Presentation:** 3 good
**Contribution:** 3 good
**Rating:** 6
**Confidence:** 4

**Summary:**

This paper focuses on the robust fairness problem, they empirically find that  the majority of the model’s predictions for samples from the worst class are biased towards classes similar to the worst class. Inspired by this, they show a theoretical result that a class is hard to learn and exhibits lower performance when it is close to other classes. They then propose a novel method, it use class-wise probability to measure the difficulty of the class. Based on this, they computing the weights which is used to reweight the loss and adjust the margin. Empirically, they show the proposed method outperforms other baselines in three different datasets.

**Strengths:**

1. The authors use experiments to show that the misclassified samples from worst class are biased towards classes similar to the worst class, and they provide a theoretical analysis on this.
2. They propose a novel method, adjusting the weights and margins based on the class-wise probability, which is used to measure the difficulty of the class.
3. The authors show experiments on three widely used datasets to show the superiority of the proposed method.

**Weaknesses:**

1. The update rule for $\mathcal{W}$ may result in negative values in some cases. For instance, when the performance of a class consistently remains the best, this issue arises. The authors do not address this problem or discuss its implications. If $\mathcal{W}$ becomes negative, it would lead to negative robust radii ($\mathcal{W}_y\epsilon$), which is unreasonable.
2. The update process for $\mathcal{W}$ appears to be time-inefficient, particularly when dealing with datasets that contain a large number of classes. It is suggested that the authors provide information on the time costs of the proposed method compared to other baselines.

**Questions:**

1. What is the theoretical and empirical range of $\mathcal{W}$? A clarification of the possible values for $\mathcal{W}$ is necessary.
2. Does the proposed method require more time for experimentation compared to other baselines?
3. What distinguishes the proposed method from other baselines, considering that these methods also adjust weights? Can the authors elaborate on the differences in the weight curves between the proposed method and the baselines?

---

> ### Author Response · Authors · 2023-11-17
> **Response to Reviewer n9NT (1/2)**
>
> Thank you for reviewing our manuscript and indicating some questionable parts in our paper, especially for our method, DAFA. We would like to address your concerns and questions.
> \
> \
> **Q**: **The update rule for $\mathcal{W}$ may result in negative values in some cases. For instance, when the performance of a class consistently remains the best, this issue arises. The authors do not address this problem or discuss its implications. If $\mathcal{W}$ becomes negative, it would lead to negative robust radii ($\mathcal{W}\_y\epsilon$), which is unreasonable.**
>
>
> **A**: Theoretically, the value of $\mathcal{W}$ can be negative. This occurs in the most extreme case, where in Eq. 7, $i$ represents the easiest class, and all other classes $j$ are almost certain to be classified as class $i$, resulting in $\min\mathcal{W}\_i \approx  \mathcal{W}\_{i,0} - 9$ for a 10-class classification. The maximum value of $\mathcal{W}\_i$ is achieved when class $i$ is the most challenging class, with $p_i(i)=0$ and $\sum_{j} p_i(j)=1$, yielding $\max \mathcal{W}\_i = \mathcal{W}\_{i,0} + 1$. To address this, we scale $\mathcal{W}$ for each dataset using the $\lambda$ value in Eq. 8, ensuring that $\mathcal{W}$ is adjusted. Additionally, we apply clipping to ensure $\mathcal{W}$ remains greater than 0 ($\mathcal{W} =$ maximum$(\mathcal{W}, K)$ where $K > 0$). In our experiments, we selected $\lambda$ values of 1.0 for the CIFAR-10 dataset and 1.5 for the CIFAR-100 and STL-10 datasets, resulting in $\mathcal{W}$ values around 1.0. The values of $\mathcal{W}$ for the CIFAR-10 dataset can be found in Fig. 7a of the paper.
> \
> \
> Furthermore, in our proposed method, we can obtain $\mathcal{W}$ in a direction that does not lead to negative values. In our approach, we restrict the learning of easy classes and promote the learning of hard classes using hard classes as a reference. This might cause the $\mathcal{W}$ values for easy classes to be negative. However, irrespective of which class between $i$ and $j$ is relatively easy or hard, if $i$ and $j$ form a similar class pair, the probability of class $i$ being classified as $j$ and the probability of class $j$ being classified as $i$ both yield high values. Conversely, if they are dissimilar, both probabilities are low. Therefore, instead of using $p_i(j)$ with hard classes as a reference, we can use $p_j(i)$ with easy classes as a reference. Then, Eq. 7 is modified as follows:
> $$
> \mathcal{W}\_{i}=\mathcal{W}\_{i,0}+\sum_{j \neq i} 1[p_i(i) < p_j(j)] \cdot p_j(i) - 1[p_i(i) > p_j(j)] \cdot p_i(j)
> $$
> Assuming the most extreme case, let $i$ be the easiest class, $p_i(i) >0$ and $\sum_{j \neq i} p_i(j) < 1$. Consequently, $\min \mathcal{W}\_i > \mathcal{W}\_{i,0} - 1$. Therefore, when $\mathcal{W}\_{i,0}=1$, $\mathcal{W}\_i$ is always greater than 0. Applying this to Eq. 8, as probabilities always fall within the range of 0 to 1, the range of $\mathcal{W}\_i$ calculated in Eq. 8 is similar to that in Eq. 7 when $\lambda=1$. Since our paper developed the argument based on hard classes, we applied the method using hard classes as a reference in the main paper.
> \
> \
> Additionally, we conduct experiments to verify the effectiveness of our method when referencing easy classes instead of hard classes. The results, including the calculated $\mathcal{W}$ values and the learning outcomes, for the CIFAR-10 dataset are presented. Although there are slight differences in the values, the ranking and trends of $\mathcal{W}$ sizes are consistent. Furthermore, experimental results indicate that clean, average robust, and worst robust accuracy are all similar to the original results.
> |  Class (the value of $\mathcal{W}$) | Plane | Car | Bird | Cat | Deer | Dog | Horse | Frog | Ship | Truck |
> |:--------:|:--------:|:--------:|:--------:|:--------:|:--------:|:--------:|:--------:|:--------:|:--------:|:--------:|
> | hard class reference | 0.978 | 0.810 | 1.078 | 1.224 | 1.163 | 1.005 | 0.924 | 0.919 | 0.851 | 1.047 |
> | easy class reference | 0.979 | 0.835 | 1.116 | 1.273 | 1.200 | 1.019 | 0.943 | 0.943 | 0.865 | 1.062 |
> ||||||||||||
>
> |  Model | Clean Avg. | Clean Worst | $\rho_{nat}$ | Robust Avg. | Robust Worst | $\rho_{rob}$ |
> |:--------|:--------:|:--------:|:--------:|:--------:|:--------:|:--------:|
> |  TRADES |82.04 $\pm$ 0.10 | 65.70 $\pm$ 1.32 | - | 49.80 $\pm$ 0.18 | 21.31 $\pm$ 1.19 | - |
> |  TRADES + DAFA (hard class reference) | 81.63 $\pm$ 0.08 | 67.90 $\pm$ 1.25 | 0.03 | 49.05 $\pm$ 0.14 | 30.53 $\pm$ 1.04 | 0.42 |
> | TRADES + DAFA (easy class reference) | 81.71 $\pm$ 0.08 | 67.28 $\pm$ 1.48 | 0.02 | 49.08 $\pm$ 0.21 | 30.22 $\pm$ 0.82 | 0.41 |
> ||||||||||
>
> Consequently, we conclude that adjusting the scale parameter $\lambda$ and applying clipping to prevent negative values effectively prevent $\mathcal{W}$ from becoming negative. Moreover, it can also be complemented by referencing easy classes, as shown, to achieve the same result.

---

> ### Author Response · Authors · 2023-11-17
> **Response to Reviewer n9NT (2/2)**
>
> **Q**: **The update process for $\mathcal{W}$ appears to be time-inefficient, particularly when dealing with datasets that contain a large number of classes. It is suggested that the authors provide information on the time costs of the proposed method compared to other baselines.**
> \
> **A**: Our proposed method is not an adaptive approach where the class weight $\mathcal{W}$ changes over time. As detailed in Appendix A, we calculate the class weight only once after the warm-up epoch and then fix it for the subsequent training. In this process, we store the values computed during the training, which are later utilized for calculating $\mathcal{W}$. Therefore, apart from the memory (class-wise probability $p$ in Algorithm 1 of Appendix A) needed to store probabilities, there is almost no additional cost for calculating $\mathcal{W}$ since we do not separately compute the probabilities for calculating $\mathcal{W}$. Thus, the additional time cost is limited to the single computation of "$\mathcal{W}$" throughout the entire training process. For a 10-class dataset, this incurs an additional time cost of less than 1 second. Therefore, our method can be considered to have the same training time as the baseline algorithms TRADES.
> \
> \
> **Q**: **What is the theoretical and empirical range of $\mathcal{W}$? A clarification of the possible values for $\mathcal{W}$ is necessary.**
> \
> **A**:The range of $\mathcal{W}$ is $\mathcal{W}\_{0}-9<\mathcal{W} \leq \mathcal{W}\_{0}+1$ when hard classes are reference, and $\mathcal{W}\_{0}-1<\mathcal{W} \leq \mathcal{W}\_{0}+9$ when easy classes are reference. The empirical range is not constrained due to $\lambda$, but setting $\lambda=1.0$ does not significantly deviate the value of $\mathcal{W}$ from around 1. Therefore, the choice of the value of $\lambda$ is not a challenging task.
> \
> \
> **Q**: **Does the proposed method require more time for experimentation compared to other baselines?**
> \
> **A**: As mentioned above, our method incurs almost negligible additional time cost when compared to the baseline training algorithm (TRADES). Unlike our approach, other methods (FRL, CFA, BAT, and WAT) are expected to have some level of additional time cost due to the need to calculate parameters, such as class weights, at each epoch.
> \
> \
> **Q**: **What distinguishes the proposed method from other baselines, considering that these methods also adjust weights? Can the authors elaborate on the differences in the weight curves between the proposed method and the baselines?**
> \
> **A**: As the reviewer correctly understands, most robust fairness improvement methods use class weights. Each method employs a distinct approach to calculating class weights, and our method differs from existing approaches by incorporating the concept of class distance to account for class similarity when calculating class weights. Additionally, as mentioned earlier, our method is not an adaptive approach; it calculates the weights once and then fixes them. Therefore, instead of examining the weight curve, we will compare the class weights with other methods at each best checkpoint. The table below illustrates the class weights for each method on the CIFAR-10 dataset, based on the weights applied to cross-entropy loss. These class weights were measured in the settings of each respective method's code. Since BAT is not a method that utilizes class weights, we have excluded it from consideration.
> |  Class | Plane | Car | Bird | Cat | Deer | Dog | Horse | Frog | Ship | Truck |
> |:--------|:--------:|:--------:|:--------:|:--------:|:--------:|:--------:|:--------:|:--------:|:--------:|:--------:|
> | FRL | 0.09	|0.09	|0.12	|0.14	|0.09	|0.12	|0.09	|0.09	|0.09	|0.09|
> | WAT |0.19	|0.09	|1.59	|2.10	|1.45	|1.69	|1.18	|0.73	|0.09	|0.32|
> |CFA | 0.13	|0.12	|0.15	|0.15	|0.14	|0.14	|0.13	|0.13	|0.12	|0.12|
> |DAFA (ours)| 0.98	|0.84|	1.12|	1.27|	1.20|	1.02|	0.94|	0.94|	0.87|	1.06|
> ||||||||||||
>
>  Since the loss scales of each method differ, the table below compares them by dividing with the median value.
> |  Class | Plane | Car | Bird | Cat | Deer | Dog | Horse | Frog | Ship | Truck |
> |:--------|:--------:|:--------:|:--------:|:--------:|:--------:|:--------:|:--------:|:--------:|:--------:|:--------:|
> | FRL | 1.00	|1.00	|1.37	|1.63|	1.00|	1.39|	1.00|	1.00|	1.00|	1.00|
> | WAT | 0.19	|0.09	|1.67|	2.20|	1.51|	1.76|	1.23|	0.77|	0.10|	0.33|
> |CFA | 0.98|	0.92|	1.17|	1.18|	1.10|	1.10|	1.01|	0.99|	0.94|	0.95|
> |DAFA (ours)| 0.98|	0.84|	1.12|	1.27|	1.20|	1.02|	0.94|	0.94|	0.87|	1.06|
> ||||||||||||
>
> Upon comparing the results, we observe a consistent trend across methods of assigning lower weights to easy classes (non-animal) and higher weights to challenging classes (animal). However, the specific values vary among the different methods. Our method exhibits a distinctive characteristic of assigning relatively lower weights to classes like dog or horse, which is similar to the worst class (cat), than truck, which is easy but dissimilar class to cat.

---

> > ### Comment · Reviewer_n9NT · 2023-11-22
> >
> > Thanks for your response. The authors point out that the additional time cost of the proposed method is limited, and they agree with the possible negative values of $\mathcal{W}$ in theory. They suggest that this can be avoided in practice by adjusting the hyperparameter $\lambda$. I suggest that the authors include a full discussion of the possibility of negative values in the manuscript. As a result. I believe this paper provides a novel insight, and I will raise my rate accordingly.

---

> > > ### Author Response · Authors · 2023-11-22
> > >
> > > Dear reviewer n9NT
> > >
> > > We appreciate once again the suggestions and comments related to our paper. Addressing the points raised and suggestions made, we delved into aspects that we had not considered, and in this process, we were able to strengthen and enhance the weaknesses of our paper. As the reviewer mentioned, we will organize the discussion and add them to the final version.

---

> ### Author Response · Authors · 2023-11-21
>
> Dear reviewer n9NT
>
> Thank you again for your insightful comments and suggestions!  We are sincerely thankful for your effort in reviewing our manuscript. As a gentle reminder, since there are only very few days left, we sincerely look forward to your follow-up response.
>
> we have submitted a rebuttal and a revised manuscript to address your mentioned concerns. We are happy to provide additional answers to illustrate the strength of our paper. In our previous responses, we carefully read your comments and made detailed responses summarized below:
>
> - We clarified that the possibility of negative values for class weights in our methodology can be managed through parameter adjustment. Additionally, we introduced an approach to address the concern by adjusting the calculation of class weights. We demonstrated an approach, theoretically preventing class weights from becoming negative values through the adjustment of reference examples and empirically validating the effectiveness of this approach.
> - We addressed the reviewer's concern about additional time cost, explaining that our methodology is not an adaptive method like previous methods. Our methodology calculates class weights only once after the warm-up epoch. Furthermore, since our method utilizes probabilities calculated and stored during the training process, there is no need for separate probability calculations for class weight computation. Taken together, the additional time cost of our methodology is entirely negligible, and the training time is the same as that of the baseline model.
> - By comparing the class weights of our methodology with those of existing methods, we illustrated the differences from other methodologies. A notable feature is that, while existing methods typically exhibit a consistent pattern of lower class weights for easy classes and higher class weights for hard classes, the class weights found by our method represent lower class weights of classes that are similar to the hard class than those of classes that are dissimilar (e.g., dog, horse, and truck class weights in the above table), leading to the highly increased performance of the hard class. Consequently, this remarkable finding indicates that restricting the learning of classes similar to the hard class leads to a significant improvement in robust fairness.
>
> We hope that the provided new experiments and the additional discussion have convinced you of the merits of this paper. Please do not hesitate to contact us if there are additional questions.
>
> Meanwhile, we would like to thank the reviewer again for the very helpful comments. By taking them into account, it would indeed make our paper clearer and stronger.
>
> Thank you for your time and effort!
>
> Best regards, \
> Authors

---

### Official Review · Reviewer_nK3C · 2023-11-07

**Soundness:** 2 fair
**Presentation:** 2 fair
**Contribution:** 2 fair
**Rating:** 5
**Confidence:** 3

**Summary:**

In this paper, the author introduced a class-distanced method aimed at enhancing fairness in Adversarial Training. To be more specific, the methods designs the robust sample loss weights and adversarial margins based on the class similarities to penalize more for hard similar classes. The author argue that the error of hard samples are mainly caused by falsely predicted as other similar classes, rather than disimilar classes. To verify this, the author first show that for the hard class, the accuracy is much improved dramatically if it trained with classes that not similar to it. Then the author derive theoretical analysis quantitively shows that the robust errors for each class, and predict error between different classes is monotonically decreasingly correlated with the gap between different classes. The large the class gap, the smaller the predictions error. Further more,  in section 4.1, the author present "an example illustrating that improving
a hard class’s performance is more effectively achieved by moving the decision boundary (DB)
between the hard class and its neighboring class than that between the hard class and a distant class". Then empirical studies on several datasets verifies the effectiveness of the proposed methods.

**Strengths:**

Please refer the summary.
Moreover, the method proposed by the author is agnostic and is empirically validated by combining it with Trades framework and an PGD to demonstrate its advantages.

**Weaknesses:**

The paper is well written. But

 1) It rambling too much until the Page 6.

2) The author only mentioned  once that  it used class prediction probability to measure the class similarities. "Our proposed method quantifies the similarity between classes in a dataset by utilizing the model output prediction probability. " When it comes to class similarity, the first idea that came to my mind is embedding similarities between different classes. It is confusing.

**Questions:**

1) What's the "red" and "blue" arrows in Figure 1 represents?
2) Are all the figures when it comes to measure the class similarities using the average class soft max probabilities you described in section 4.2 when you proposing your methods?
3) Please proofread the paper and simplify the sentences for better comprehension such as "Allocating a larger adversarial margin to the hard class than to adjacent classes has the effect of pushing the DB away from the hard class’s center by an amount corresponding to the difference in their adversarial margins."

---

> ### Author Response · Authors · 2023-11-17
> **Response to Reviewer nK3C (1/2)**
>
> Thank you for reviewing our manuscript and indicating some confusing parts in our paper. We would like to address your concerns and questions.
> \
> \
> **Q**: **It rambling too much until the Page 6.**
> \
> **A**: If reading and comprehending our manuscript up to Page 6 was challenging, we apologize. Our intention was to address the issue of robust fairness from a novel perspective, focusing on class similarities. We incorporated various motivations and analyses into our work. These include examples from 5-class classification tasks (Fig. 2), theoretical analysis on the impact of distance between two classes in a binary classification task (Section 3.1), empirical verification (Section 3.2 and Fig. 3), a comparison between class-wise variance and class-wise distance (Section 4.1 and Fig. 4), and an illustrative example (Fig. 6) outlining the process leading to the proposed method. The abundance of content and its less straightforward nature may pose challenges in comprehension. Consequently, we agree with the reviewer's feedback, and to facilitate content organization and enhance clarity, we will streamline the content up to Page 6, focusing on essential aspects to maintain the overall flow and direction of the paper. We have added responses to other questions in the Appendix and will continue to address this.
> \
> \
> **Q**: **The author only mentioned once that it used class prediction probability to measure the class similarities. "Our proposed method quantifies the similarity between classes in a dataset by utilizing the model output prediction probability." When it comes to class similarity, the first idea that came to my mind is embedding similarities between different classes. It is confusing.**
> \
> **A**: As the reviewer is aware, class similarity is mainly measured through embedding similarity. The reason we employed prediction probability as a measure of class similarity in our methodology is based on referencing previous studies [1] that utilized prediction probability as a class similarity indicator. However, it is noteworthy that embedding similarity, as mentioned by the reviewer, can also be incorporated into our methodology. To validate its practical feasibility, we conduct experiments as outlined below.
>
> |  Model | Clean Avg. | Clean Worst | $\rho_{nat}$ | Robust Avg. | Robust Worst | $\rho_{rob}$ |
> |:--------|:--------:|:--------:|:--------:|:--------:|:--------:|:--------:|
> |  TRADES |82.04 $\pm$ 0.10 | 65.70 $\pm$ 1.32 | - | 49.80 $\pm$ 0.18 | 21.31 $\pm$ 1.19 | - |
> |  TRADES + DAFA | 81.63 $\pm$ 0.08 | 67.90 $\pm$ 1.25 | 0.03 | 49.05 $\pm$ 0.14 | 30.53 $\pm$ 1.04 | 0.42 |
> | TRADES + DAFA (embedding) | 81.87 $\pm$ 0.12 | 69.80 $\pm$ 1.35 | 0.06 | 48.92 $\pm$ 0.08 | 29.60 $\pm$ 1.27 | 0.37 |
> ||||||||||
>
> As evident from the results, utilizing embedding similarity yields outcomes similar to the conventional results obtained using prediction probability. It is observed that embedding similarity also demonstrates effectiveness in improving robust fairness, as indicated by the results.
> \
> \
> Our method focuses on enhancing robust fairness by calculating class weights using class-wise similarity. Thus, our framework is not limited to prediction probability but can accommodate various types of class-wise similarity. Therefore, leveraging embedding similarities in our method is also feasible. In the above experiments, we employed the penultimate layer output as embedding vectors (512-size vector for ResNet18). The cosine similarity between vectors for each class is computed, normalized, and subsequently applied to the DAFA framework. Let $e_i$ represent the mean embedding vector of class $i$. The similarity between class $i$ and class $j$ can be expressed as the normalized value of the cosine similarity between the two classes. In other words, if we denote $p_i(j)$ as the similarity between class $i$ and class $j$, then $p_i(j)=e_i \cdot e_j / Z$, where $Z=\sum_{k \in \{1,...,C\}} e_i \cdot e_k$. We applied the computed $p_i(j)$ to Eq. 8 in the paper and proceeded with the experiments. We chose $\lambda=0.1$ since the variance of similarity values calculated using embeddings is not significant.
> \
> \
> Drawing insights from various studies on unsupervised learning and fine-tuning [2], it is apparent that embedding similarities between different classes are intricately linked to the similarity measured by prediction probability. Consequently, we can expect that the results obtained using embedding similarity as a measure would align with those obtained using prediction probability. As expected, both the prediction probability we employed and embedding similarity function effectively as indicators of class similarities.
> \
> \
> [1] Hinton, Geoffrey, Oriol Vinyals, and Jeff Dean. "Distilling the knowledge in a neural network." arXiv preprint arXiv:1503.02531 (2015). \
> [2] Chen, Ting, et al. "A simple framework for contrastive learning of visual representations." International conference on machine learning. PMLR, 2020.

---

> ### Author Response · Authors · 2023-11-17
> **Response to Reviewer nK3C (2/2)**
>
> **Q**: **What's the "red" and "blue" arrows in Figure 1 represents?**
> \
> **A**: The red and blue arrows next to "Hard," "Easy," "Similar," and "Dissimilar" in Fig. 1 represent the facilitation or restriction of learning for the respective classes or class sets. When pointing upwards, it indicates the facilitation of learning for the associated class, whereas pointing downwards signifies a restriction of the learning for that class. The arrow size corresponds to the degree of facilitation or restriction. The color of the arrows is simply for differentiation, and the roles of the arrows are the same. Upon examining Fig. 1, the arrow for the class "cat" points upwards, denoting the facilitation of learning. This is because the cat is considered the worst class, and to improve robust fairness, learning for the cat class is promoted by restricting the learning of other classes. The conventional approach ("Hard" and "Easy") restricts hard classes less (small arrows) and easy classes more (large arrows). In contrast, our proposed method, considering the relative difficulty between classes and incorporating class-wise similarity, introduces new blue arrows alongside the existing red arrows. Similar classes to the cat receive additional downward blue arrows beyond the small red arrows, as they are more restricted compared to the conventional difficulty-based approach. Conversely, dissimilar classes to the cat receive additional upward blue arrows beyond the red arrows, as they are less restricted compared to the conventional approach. Given the complexity of conveying two different methods in a single figure, it appears confusing. To address this, we add detailed explanations in Appendix D.2 for better clarity.
> \
> \
> **Q**: **Are all the figures when it comes to measure the class similarities using the average class soft max probabilities you described in section 4.2 when you proposing your methods?**
> \
> **A**: As mentioned earlier, our method allows for the incorporation of various types of class similarity. To elaborate on the similarity utilized in each figure, in Fig. 1, which provides an overview of our method, we showcase the results of experiments applying DAFA using average class softmax probabilities. Figures in the main paper, excluding Fig. 1, illustrate analyses based on the TRADES model, and detailed explanations for these are provided in Appendix D. In Figs. 3a and 4, we employed the penultimate layer output (embedding) of the TRADES model to measure class-wise distance and variance. Although explanations are available in the Appendix, there might be considerable confusion while reading the main text. After securing space in the main paper, we will clarify which measurement was utilized in each analysis.
> \
> \
> **Q**: **Please proofread the paper and simplify the sentences for better comprehension such as "Allocating a larger adversarial margin to the hard class than to adjacent classes has the effect of pushing the DB away from the hard class’s center by an amount corresponding to the difference in their adversarial margins."**
> \
> **A**: As per the reviewer's feedback, it seems that some parts of the paper are intricately worded. We will carefully re-read the paper and identify sections that might be challenging to understand or are complex, aiming to simplify them. In particular, we acknowledge that Section 4, highlighted by the reviewer, appears to be densely written, and we will make concise modifications to the relevant portions. Referring to the provided example sentence, we've identified similar areas that need improvement. we will review the paper again, streamline the sentences, and enhance readability accordingly including the below examples.
> - Page 6: Allocating a larger adversarial margin to the hard class than to adjacent classes has the effect of pushing the DB away from the hard class’s center by an amount corresponding to the difference in their adversarial margins.
> \
> → Allocating a larger adversarial margin to the hard class than adjacent classes pushes the DB away from the hard class's center by the margin difference.
> - Page 6: To confirm that allocating a larger adversarial margin to a specific class, relative to its neighboring class, shifts the DB and enhances that class’s performance, we conduct experiments with varied adversarial margins across different classes.
> \
> → We conduct experiments with different adversarial margins across classes to verify that assigning a larger margin to a specific class, compared to its neighboring class, shifts the DB and improves the class performance.
> - Page 7: We devise and utilize a new weight allocation approach where we decrease the weight of the relatively easier class by a specific amount and increase the weight of the harder class by the same magnitude.
> \
> → We devise a new weight allocation approach: we reduce the weight of the easier class by a specific amount and simultaneously increase the weight of the harder class by the same magnitude.

---

> ### Author Response · Authors · 2023-11-21
>
> Dear reviewer nK3C
>
> Thank you again for your insightful comments and suggestions!  We are sincerely thankful for your effort in reviewing our manuscript. As a gentle reminder, since there are only very few days left, we sincerely look forward to your follow-up response.
>
> we have submitted a rebuttal and a revised manuscript to address your mentioned concerns. We are happy to provide additional answers to illustrate the strength of our paper. In our previous responses, we carefully read your comments and made detailed responses summarized below:
>
> - We provided explanations and supplements for the complex aspects pointed out by the reviewer, incorporating modifications or additions to the manuscript. Specifically, we added additional explanations for the analyses presented through figures and replaced complex sentences with clearer expressions.
> - Considering the possibility that class-wise similarity can be perceived as the embedding similarity mentioned by the reviewer, we conducted additional experiments using embedding similarity as class-wise similarity in our method. The experimental results show similar outcomes to when we utilized prediction probability as class-wise similarity, demonstrating that embedding similarity behaves similarly to class-wise similarity of prediction probability. Through this, we illustrated that our methodology is not limited to the use of prediction probability but can utilize general class-wise similarity, including embedding similarity.
>
> We hope that the provided new experiments and the additional discussion have convinced you of the merits of this paper. Please do not hesitate to contact us if there are additional questions.
>
> Meanwhile, we would like to thank the reviewer again for the very helpful comments. By taking them into account, it would indeed make our paper clearer and stronger.
>
> Thank you for your time and effort!
>
> Best regards, \
> Authors

---

### Meta-Review · Area_Chair_Ugpr · 2023-12-05

**Metareview:**

The authors address the issue of robust fairness - where heterogeneity in difficulty between classes (due to say, two classes that are similar) leads to gaps in accuracy and fairness across the classes.

Strengths: the empirical validation is pretty extensive, especially after the rebuttal, and there is additional theoretical justification of the approach. Conceptually, the method also gives a solution to an interesting problem.

Weaknesses: reviewer 73vn rightfully pointed out some issues like similarity of the theory to prior work. several reviewers had comments about the baselines lacking, though this was the subject of a number of follow ups during the rebuttal

**Justification For Why Not Higher Score:**

The authors addressed some issues (like extensiveness of baselines) but the theory isn't that novel and having extensive baselines does not justify a spotlight or oral.

**Justification For Why Not Lower Score:**

The authors addressed the substantive reasons to reject (baselines, minor rewrites for novelty)

---

### Decision · Program_Chairs · 2024-01-16

Accept (poster)